# Inherently confinable split-drive systems in *Drosophila*

Gerard Terradas [1,2], Anna B. Buchman[1], Jared B. Bennett [3], Isaiah Shriner[1], John M. Marshall [4,5], Omar S. Akbari [1] & Ethan Bier [1,2✉]

CRISPR-based gene-drive systems, which copy themselves via gene conversion mediated by the homology-directed repair (HDR) pathway, have the potential to revolutionize vector control. However, mutant alleles generated by the competing non-homologous end-joining (NHEJ) pathway, resistant to Cas9 cleavage, can interrupt the spread of gene-drive elements. We hypothesized that drives targeting genes essential for viability or reproduction also carrying recoded sequences that restore endogenous gene functionality should benefit from dominantly-acting maternal clearance of NHEJ alleles combined with recessive Mendelian culling processes. Here, we test split gene-drive (sGD) systems in *Drosophila melanogaster* that are inserted into essential genes required for viability (*rab5*, *rab11*, *prosalpha2*) or fertility (*spo11*). In single generation crosses, sGDs copy with variable efficiencies and display sex-biased transmission. In multigenerational cage trials, sGDs follow distinct drive trajectories reflecting their differential tendencies to induce target chromosome damage and/or lethal/ sterile mosaic Cas9-dependent phenotypes, leading to inherently confinable drive outcomes.

[1] Section of Cell and Developmental Biology, University of California, San Diego, La Jolla, CA, USA. [2] Tata Institute for Genetics and Society, University of California, San Diego, La Jolla, CA, USA. [3] Biophysics Graduate Group, Division of Biological Sciences, College of Letters and Science, University of California, Berkeley, CA, USA. [4] Divisions of Epidemiology and Biostatistics, School of Public Health, University of California, Berkeley, CA, USA. [5] Innovative Genomics Institute, Berkeley, CA, USA. ✉email: ebier@ucsd.edu

CRISPR-based gene-drive systems offer new potential strategies for controlling pest populations[1] or disease-transmitting vectors[2–5]. So-called efficient "low-threshold" CRISPR gene-drive systems (capable of spreading from low-seeding frequencies) rely on a cut-and-repair mechanism, wherein Cas9 produces a double-stranded break (DSB) in the DNA followed by homology-directed repair (HDR) using the homologous chromosome as a template and copying of the genetic element into the break on the Cas9-cleaved chromosome[6,7]. However, the competing non-homologous end-joining (NHEJ) pathway can generate alleles resistant to Cas9 cleavage[8,9] that interrupt spread of the desired genetic element[10], particularly if the cleavage-resistant allele carries less of a fitness cost than the driving allele. NHEJ-induced alleles consisting primarily of short insertions or deletions (indels) result in amino acid substitutions or frameshifts that modify the gRNA target sequence, preventing further Cas9 cleavage[11]. Non-functional drive-resistant loss-of-function (LOF) mutations, typically the most abundant NHEJ products, can delay complete drive introduction since they are eliminated only gradually by negative selection when homozygous[10]. Another important consideration is to generate as few, if any, *in frame* functional indels, since such cleavage resistant alleles could compete with the drive element. This latter concern is particularly relevant to suppression drive systems that they would rapidly outcompete, but also could impede the spread of modification systems if such functional alleles were numerous and/or were more fit than the drive allele.

Several strategies have been proposed to reduce the incidence or effects of cleavage-resistant alleles. One approach is to employ tightly-regulated germline-specific promoters to avoid somatic expression of Cas9 that can lead to NHEJ-induced repair events[9,10,12]. Some germline promoters are quite specific, at least in certain genomic contexts (e.g., *nanos*[13] or *zpg*[14]), while others are leaky (e.g., *vasa*[15,16]). Another tactic is to identify highly-conserved genomic targets as cleavage sites such that all NHEJ mutants suffer severe fitness costs[14,17–19].

Recently, it has been suggested that drives targeting conserved genes essential for survival or fertility that also carry a recoded cDNA restoring endogenous gene activity would benefit from two forms of positive selection in populations[2,17,19–21]. The first advantage results from gradual Mendelian culling of recessive deleterious NHEJ alleles. The second, a more rapid process, depends on an actively dominant phenomenon referred to as lethal/sterile mosaicism, in which maternal deposition of Cas9/gRNA complexes in the embryo mutates the paternal allele in a mosaic fashion[2,22]. If progeny inherit the recoded drive, they are protected from this maternal effect since they carry one unassailable functional allele. Individuals inheriting a non-functional NHEJ allele, however, will either die or be infertile if the targeted gene is required broadly in many cells of the organism.

Two main types of CRISPR-based gene drives, or "active genetic" elements, have been designed and tested. The first, full gene drives (fGD)[2,3,6,15], have linked sources of Cas9 and gRNA that are inserted together as one unit at a single genomic site. The second, split gene drives (sGD)[5,16,23,24], carry a gRNA-only cassette capable of copying when combined with a "static" source of Cas9 inherited in a Mendelian fashion and located at a second locus. Split systems also have been deemed safer for laboratory research[25,26] and more amenable for localized release purposes[5,23,27].

In this study, we design and test several sGDs in *Drosophila melanogaster* with various genetic parameters and strategies either to limit or extend sGD drive potential. We demonstrate that resistance can be largely overcome if the drive allele carries an efficient gRNA and recoded target gene sequences while generating few functional drive-resistant NHEJ alleles. sGDs inserted into target genes essential for survival or fertility were efficiently transmitted through females, but generally to a lesser degree through males. Genomic context (Cas9 promoter, chromosomal location) also contributed to gene-drive efficiency and rates of NHEJ mutagenesis. In multigenerational cage studies, sGDs benefited from the dual effect of a dominant maternal process (lethal/sterile-mosaicism) and classic gradual zygotic Mendelian negative selection. These advantages were amplified in the case of sGDs inserted into loci that demonstrate Cas9-dependent fitness costs. In such cases, sGDs behave as self-limiting and inherently confinable drive systems by virtue of imposing a fitness cost when Cas9 is co-inherited with the drive element.

## Results

**Design of split gene-drive (sGD) elements in autosomal recessive lethal loci.** We tested whether super-Mendelian transmission efficiencies (i.e., >50% expected by Mendelian inheritance) could be achieved in bipartite gene-drive systems inserted into essential gene targets on the II or III chromosomes (Fig. 1a, d) targeting conserved DNA coding sequences of recessive lethal (rab5, rab11, prosalpha2) or sterile (mei-W68[28], the *D. melanogaster* ortholog of *spo11* - referred from here on as *spo11*) genes. These elements, referred to hereafter as split gene-drives (sGDs), contain a gRNA selected to target sequences at, or close to, critical amino acids (e.g., catalytic centers or membrane-tethering sequences) of each gene of interest to minimize the potential generation of functional NHEJ events; a recoded cDNA portion of the gene that is seamlessly fused to endogenous coding sequences to restore its functionality upon successful insertion; a 3xP3-tdTomato dominant marker, and a partial inactive fragment of Opie2-eGFP (designed to permit future conversion of each sGD into fGD[29]). All the aforementioned cargo is flanked by 1 kb homology arms to the gRNA cut site to support HDR-mediated integration of the cassette into the genome (Fig. 1a). The sGDs were targeted to loci selected based on various criteria including: function and evolutionary conservation (all loci); propitious localization features (rab5, rab11[30,31]); broad cellular and/or dosage sensitive requirements (prosalpha2[32]). If possible, gRNAs targeted locations near the carboxy-terminus to minimize the extent of recoding necessary (e.g., few terminal amino acids of rab5 and rab11). Also, by choosing essential target genes, more prevalent classes of loss-of-function (LOF) alleles will incur severe fitness costs (e.g., lethality or sterility) resulting from homozygosis or from the potent filtering phenomenon of dominantly-acting somatic lethal/sterile mosaicism[2,22] (Supplementary Fig. 1).

**sGDs display super-Mendelian inheritance.** We first tested the frequency at which sGDs copied onto a wildtype (WT) chromosome in combination with a static source of Cas9 provided *in trans* from the X, II or III chromosomes (Fig. 1b). All sGD and Cas9 elements were inherited at Mendelian frequencies as separate elements. However, when combined with Cas9 sources inserted in different chromosomes and driven by different promoters (*vasa* - vCas9 or *nanos* - nCas9), sGD cassettes copied to the homologous chromosome leading to super-Mendelian transmission (Fig. 2). $G_0$ sGD homozygous males or virgin females were crossed to Cas9-bearing lines to obtain trans-heterozygous $F_1$ individuals carrying both transgenes (sGD; Cas9), which we refer to as 'master females' or 'master males'. Master males or females (virgin) were single-pair mated to WT individuals of the opposite sex, and resulting $F_2$ progeny were assessed for efficiency of sGD copying by scoring the prevalence of red-eye (3xP3-tdTomato, for the sGD element), and

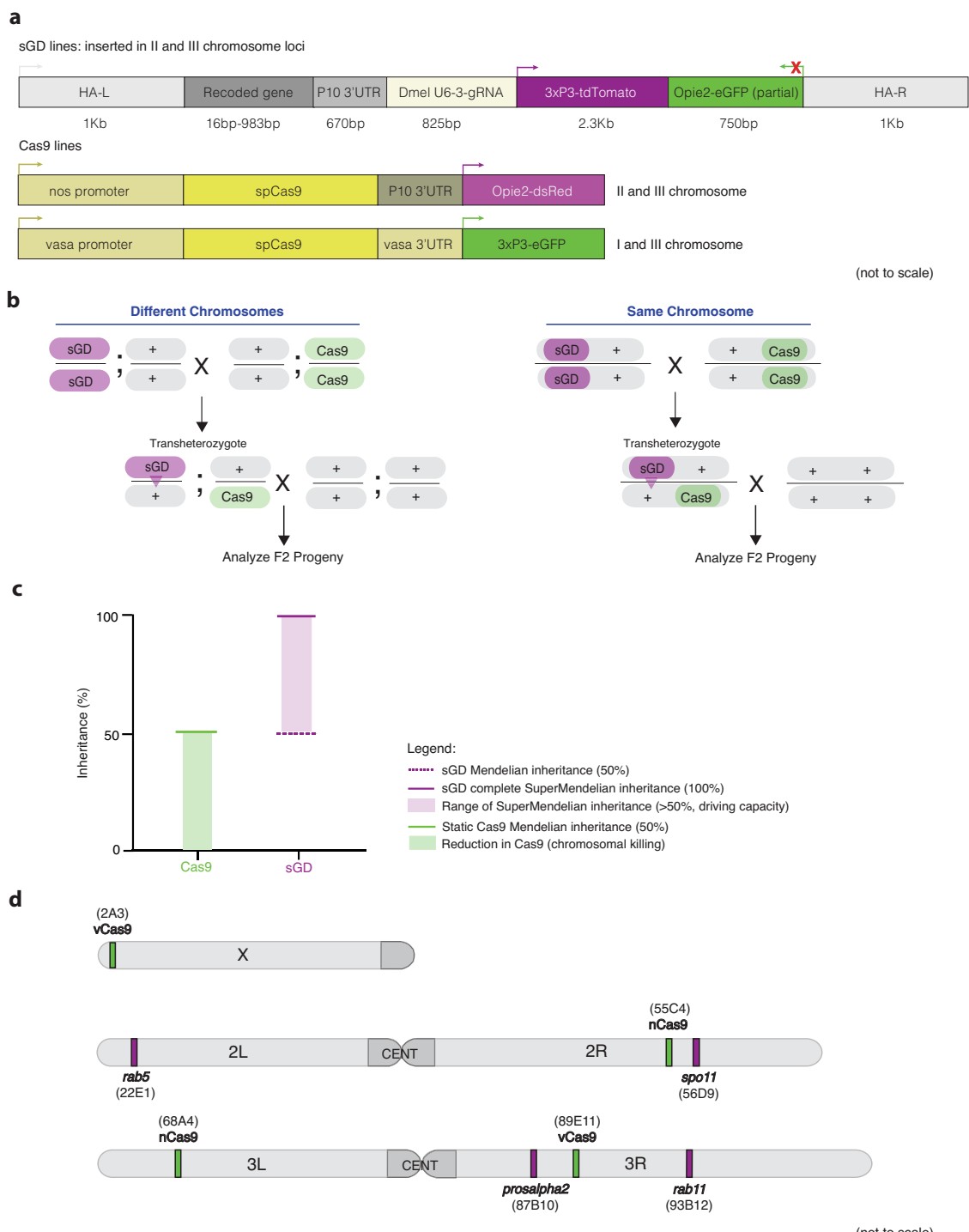

**Fig. 1 Experimental design of the split gene-drive system in essential loci. a** Schematic of the genetic constructs engineered and tested in the study. All constructs contain a recoded cDNA of the target gene that restores its functionality upon insertion of the transgene, a specific gRNA, and expression of 3xP3-tdTomato. Static Cas9 lines encode a *nanos* or *vasa*-driven Cas9 and a selectable marker, Opie2-dsRed or 3xP3-GFP, respectively. **b** Outline of the genetic cross schemes used to demonstrate the driving efficiency of each sGD, comparing systems where the sGD, Cas9 and wildtype (WT, +) are located in the same (right panel) or different chromosomes (left panel). $F_1$ trans-heterozygotes (carriers of both Cas9 and sGD *in trans*) were singly crossed to WT individuals to assess germline transmission rates by scoring % of the fluorescence markers in $F_2$ progeny. The conversion event at the sGD locus is shown with a triangle in $F_1$ individuals. **c** Overview of how data is plotted throughout the paper. $F_1$ germline inheritance is plotted in two independent columns, one that refers to the static Cas9, which should be inherited at Mendelian ratios since it does not have driving capacity, and a second column that displays the biased inheritance of the sGD transgene. Graph contains no empirical data. **d** Chromosomal location and insertion sites of all sGD and static Cas9 transgenes in the *Drosophila melanogaster* genome.

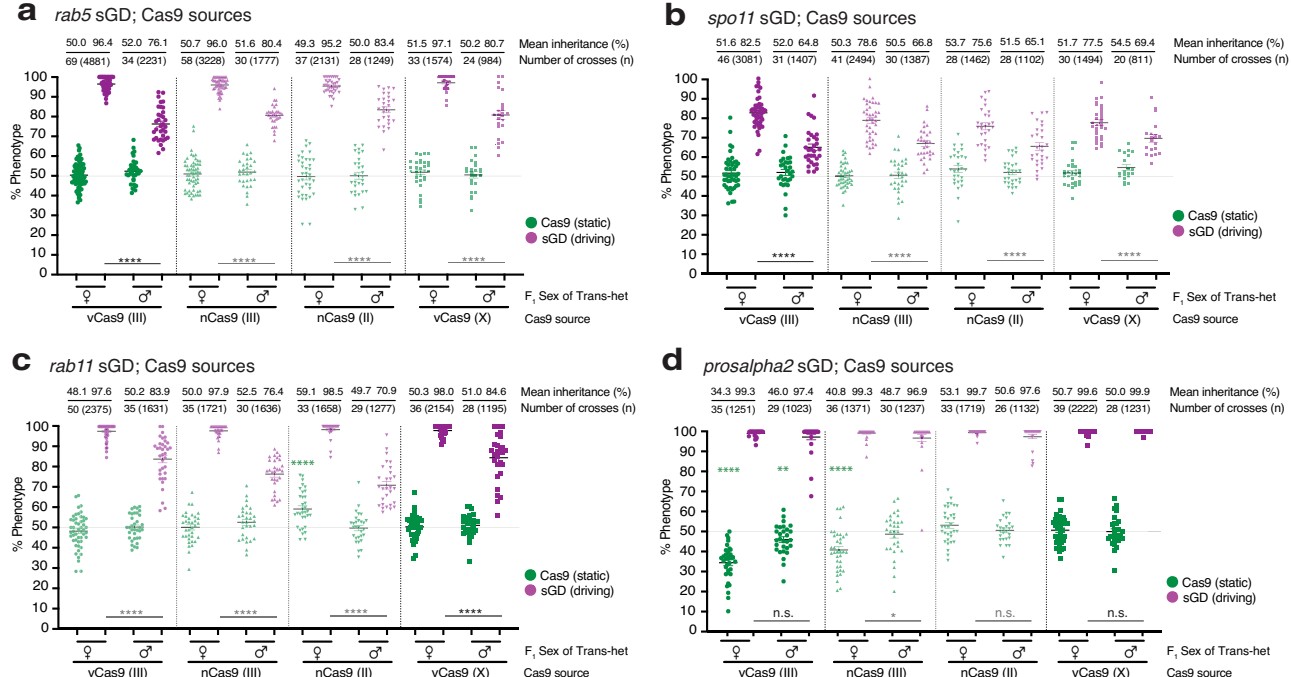

**Fig. 2 sGD elements display different super-Mendelian inheritance patterns depending on the trans-heterozygote progenitor sex.** Genetic crosses performed using the sGD transgenes in combination with *vasa* or *nanos*-driven Cas9 lines located in the first (X), second (II) or third (III) chromosome. Graph contains data for **a** *rab5*; **b** *spo11*; **c** *rab11*; and **d** *prosalpha2* sGDs. Single $F_1$ germline conversion was assessed by scoring the markers for both transgenes in the $F_2$ progeny. Inheritance of Cas9 and sGD is depicted using green and purple dots, respectively. Each single cross is shown as a single data point. Values for inheritance mean, number of crosses (N) and individuals scored (n) are shown atop of the graph in line with each respective dataset. Sex of the parental ($F_1$) trans-heterozygote is indicated in the X-axis. sGD-Cas9 combinations depicted in bold represent pairings that were progressed to cage trials and further genotypic studies. Error bars represent mean values ± SEM. Stars represent statistical significance (****$p < 0.0001$; ***$p < 0.001$; **$p < 0.01$; *$p < 0.05$) for $F_1$ male-to-female sGD-copying differences (black, two-sided *t*-test) and to detect Cas9 being inherited below Mendelian frequencies (green, $\chi^2$). Raw phenotypical data is provided as "Supplementary Data 1".

inheritance of Cas9-linked fluorescent markers (green eyes: 3XP3-eGFP, for vCas9 or red bodies: Opie2-DsRed, for nCas9).

$F_2$ progeny displayed super-Mendelian inheritance of sGDs ranging from 64.8% to 99.9%, depending on insertion locus, source of Cas9, and sex of the $F_1$ trans-heterozygote parent (Fig. 2). We detected no differential drive performance between Cas9 being contributed by $G_0$ males versus females. However, sex-dependent differences were observed for transmission of the *rab5*, *rab11*, or *spo11* sGDs from $F_1$ parents to $F_2$ progeny. When passed through $F_1$ master females, *rab* sGDs were inherited with high efficiency (95.2–97.1% for *rab5*; 97.6–98.5% for *rab11*, Fig. 2a and c) for all sGD-Cas9 combinations, while transmission of the *spo11* sGD was somewhat lower (75.6–82.5%, Fig. 2b). In contrast, when transmitted from $F_1$ master males, inheritance of these sGDs dropped by ~15% (76.1–83.4% for *rab5*; 70.9–84.6% for *rab11*; 64.8–69.4% for *spo11*, Fig. 2) relative to rates observed with $F_1$ master females. These trends are readily apparent when data are compiled for each locus across Cas9 sources (*rab5* sGD (*t*-test): $U = 1010$, $n_1 = 197$, $n_2 = 116$, $p < 0.0001$; *rab11* sGD (*t*-test): $U = 1052$, $n_1 = 154$, $n_2 = 122$, $p < 0.0001$; *spo11* sGD (*t*-test): $U = 2613$, $n_1 = 145$, $n_2 = 109$, $p < 0.0001$, Supplementary Fig. 2). An exception to this sex-biased drive was the *prosalpha2* sGD, which was transmitted with similarly high efficiency through $F_1$ master females (99.3–99.7%, Fig. 2d) and $F_1$ master males (97.4–99.9%, Fig. 2d; *prosalpha2* sGD (*t*-test): $U = 7640$, $n_1 = 143$, $n_2 = 113$, $p = 0.256$, Supplementary Fig. 2).

Several observations and trends can be extracted from the data summarized above. First, observed locus-specific variation in copying efficiencies most likely depends on several factors including the genomic context of the insertion locus, gRNA

sequence, the strength (e.g., vCas9-III > nCas9-III ~ vCas9-X > nCas9-II) and degree of germline specificity (nCas9 > vCas9) of the Cas9 source (Supplementary Figs. 1 and 2). Second, the essential nature of the target gene is highly relevant in this context: genes that are broadly required in the organisms for viability (*rab5*, *rab11*, *prosalpha2*) may display varying degrees of lethal mosaicism, depending on whether their activities are required in many or few cells or in a cell-autonomous fashion. These results suggest that a stringent requirement for locus function results in more complete and immediate elimination of LOF alleles (*prosalpha2* > *rab5*), whereas for other genes (*rab11*) individuals heterozygous for LOF alleles may suffer a less significant fitness reduction. Targets essential for reproduction, such as *spo11*, will generate sterile mosaic progeny that survive but are infertile, and most NHEJ events will not be immediately eliminated from the population, but will lead to dominant sterility in subsequent generations.

As summarized above, the highly-efficient *prosalpha2* sGD was an exception to the sex-biased pattern of transmission. Crosses with the *prosalpha2* sGD employing the 3rd chromosome vCas9 source, which also served as a marker for the receiver chromosome due to its close linkage to the sGD cleavage site (~2 cM, Fig. 1d), revealed another behavior unique to this sGD in which sub-Mendelian inheritance of the GFP-marked Cas9 element was observed (Fig. 2d). In all other instances, recombination between the Cas9 source and the sGD (in females) or independent chromosome assortment precluded discrimination between donor versus receiver chromosomes. Allele-specific analysis in the case of the 3rd chromosome vCas9 source revealed that the receiver chromosome was being transmitted at

significantly lower rates than expected in both sexes (35–40% versus expected 50%, chi-square goodness-of-fit—in females: $\chi^2$ (1, $N = 1251$) = 104.2, $p < 0.0001$; in males: $\chi^2$ (1, $N = 1023$) = 7.1, $p < 0.05$, Fig. 2d). In principle, this observation could result from the copying process damaging the receiver chromosome and resulting in the homolog being inadequately repaired and lost prior to fertilization or shortly thereafter. Alternatively, the chromosome-III sources of Cas9 (both vCas9 and nCas9) may be stronger (e.g., higher levels or more timely delivery of Cas9) leading to individuals that carry both the sGD transgene and Cas9 source being less viable than those bearing the drive element alone. Selective sub-Mendelian transmission of Cas9 in master female crosses employing nCas9-III (chi-square goodness-of-fit: $\chi^2$ (1, $N = 1371$) = 44.5, $p < 0.0001$, Fig. 2d) is consistent with a greater relative activity of this Cas9 source, since it is unlinked to the *prosalpha2* sGD locus and should freely recombine with any chromosomal damage incurred at the gRNA cut site. In crosses of the *prosalpha2* sGD to Cas9 sources located on the 2nd or the X chromosomes we did not observe such deficits in GFP inheritance, suggesting that these Cas9 sources may express the Cas9 transgene at lower levels or in a more germline specific fashion than those delivered by the 3rd chromosome sources of Cas9.

We also evaluated the sGD systems for the phenomenon of 'shadow drive', in which maternally-deposited Cas9 in the egg biases the inheritance of a transgene for one extra generation even if the Cas9 element is not transmitted to $F_2$ females[16,22]. We crossed $F_2$ sGD+/Cas9− virgin females (that only carried maternally-deposited Cas9 protein) to WT males and scored for presence of the eye-tdTomato marker in $F_3$ individuals. In the absence of shadow drive, $F_3$ progeny should display Mendelian inheritance of the marker. However, progeny of sGD+/Cas9− $F_2$ females exhibited modest levels of super-Mendelian transmission of sGDs, which was most pronounced for the *rab5* sGD+ vCas9-III (64.8 ± 11.9% = ~30% conversion, Supplementary Fig. 3). We also performed the reciprocal cross of $F_2$ sGD+/Cas9− males to wildtype virgin females. Consistent with little appreciable paternal Cas9 protein being transmitted in the sperm of such $F_2$ males, sGDs were inherited at expected Mendelian rates in their $F_3$ progeny (~50%, = 0% conversion, Supplementary Fig. 3).

**Resistant alleles do not impede sGD performance.** Because lethal mosaicism is hypothesized to dominantly eliminate LOF NHEJ mutations[2,22], we analyzed production of these alleles in single-generation crosses as well as their predicted elimination in multigenerational cage studies. In single-generation experiments (Fig. 2), we recovered individual non-fluorescent male and female $F_2$ individuals (i.e., flies lacking the sGD dominant marker) that carried potential NHEJ alleles from one parent and a WT allele on the homologous chromosome provided by the other parent. Knowing the WT sequence, we were able to distinguish WT Sanger sequencing chromatogram reads from potential indel alleles. This analysis revealed the production of indels for all genes (including LOF frameshift and *in frame* mutants) as well as intact WT alleles (Fig. 3). For the *rab5* sGD, *in frame* NHEJ alleles accounted for 34.5% of all NHEJ events (Fig. 3a), equating to 1.2% of total alleles in $F_2$ progeny. Rab proteins contain carboxy-terminal prenylation sites, consisting of two cysteine residues that form disulfide bonds essential for their localization to vesicular membranes[33]. Some *in frame* NHEJ events mutated one or both of these two cysteines since, by design, the gRNA cleavage sites were chosen to be very near these codons. Consistent with the role of Rab protein tail residues in forming essential disulfide bonds, no cysteine-altering NHEJ alleles were recovered (even as balanced heterozygotes) among 12 tested balanced $F_2$ alleles

despite 31% of the individuals carrying frameshift mutations that presumably disrupt disulfide bond formation and prevent protein localization to vesicle membranes.

Similar categories of mutant alleles were recovered in the *spo11* and *rab11* sGD crosses. The *spo11* sGD generated few frameshift mutations (Fig. 3b), and *in frame* mutations accounted for 7.7% of total $F_2$ alleles. Since the gRNA cut site for this drive targets a sequence adjacent to critical catalytic residues required for its topoisomerase activity, many of these *in frame* mutants are likely to be sterile when homozygous, as would be frameshift mutations leading to truncated non-functional proteins[28]. *Rab11* has a similar prenylated 3' end structure as *rab5*[34], with dual cysteines located at the tail end of the protein. Interestingly, fewer mutations altered these cysteine residues, even when in a heterozygous condition (13%, Fig. 3c). However, a higher percentage (63%, Fig. 3c) of the non-fluorescent individuals harbored *in frame* NHEJ alleles, accounting for 1.2% of the overall target alleles, similar to the those recovered for the *rab5* sGD. This marked skew in recovering predominantly *in frame* alleles is consistent with LOF frameshift alleles being created and then immediately eliminated by lethal mosaicism, and validates the strategy of selecting gRNA cut sites in functionally critical domains. In the case of the *prosalpha2* sGD, we did not detect a single WT allele (Fig. 3d) among non-fluorescent $F_2$ progeny, suggesting Cas9-mediated ~100% cleavage at this locus. The very low rate of total recovered NHEJ events, with *in frame* NHEJ events accounting for only 0.16% of the total alleles, correlates with the high transmission frequencies observed in single pair crosses. This high degree of sGD transmission may result from particularly efficient copying at the *prosalpha2* locus and/or individuals carrying NHEJ-induced LOF alleles or damaged chromosomes rarely surviving (e.g., due to strong lethal mosaicism).

**sGD exhibit differing degrees of drive and Cas9 depletion in multigenerational cages.** We next assessed the potential of the various sGDs to mediate population modification by testing their performance in small laboratory population cages. $G_0$ drive-competent master males and master female virgins, heterozygous for both the sGD and vCas9 elements (sGD/+; vCas9/+), were combined with male and virgin female WT flies at a seeding ratio of 1:3 sGD/+; vCas9/+ to WT. Since the drive-bearing flies were heterozygous for the transgenes, this initial introduction corresponds to 12.5% sGD and Cas9 relative to total alleles. Except for the *prosalpha2* sGD trials in which vCas9-III is closely linked to the sGD insertion locus, the sGD and Cas9 elements were unlinked and could segregate independently. At each generation, we scored half of the progeny in a cage for prevalence of both the tdTomato+ (sGD) and eGFP+ (vCas9) elements based on their fluorescent phenotypes. The other half was used to seed the next generation.

The *rab5* sGD (Fig. 4a) and *spo11* sGD (Fig. 4b) cage trials were conducted for 20 generations using a vCas9-III source. Both drives achieved 80-90% introduction by $G_{5-7}$, and remained stable with little variation in overall population size or bias in male to female ratios over the remaining generations (Supplementary Fig. 4). The long plateau phases observed in all replicates, extending until $G_{20}$, most likely reflect an equilibrium being attained between the sGD and a stable percentage of drive-resistant functional NHEJ alleles (Fig. 3e and f). The NHEJ allele dynamics differ substantially between sGDs. A class of non-functional resistant allele was observed at $G_2$ for the *rab5* sGD (Fig. 3e) and then disappeared ($G_4$ and $G_8$). A few residual *in frame* cysteine-containing, and potentially functional, resistant alleles fixed in the population. In contrast, for the *spo11* sGD

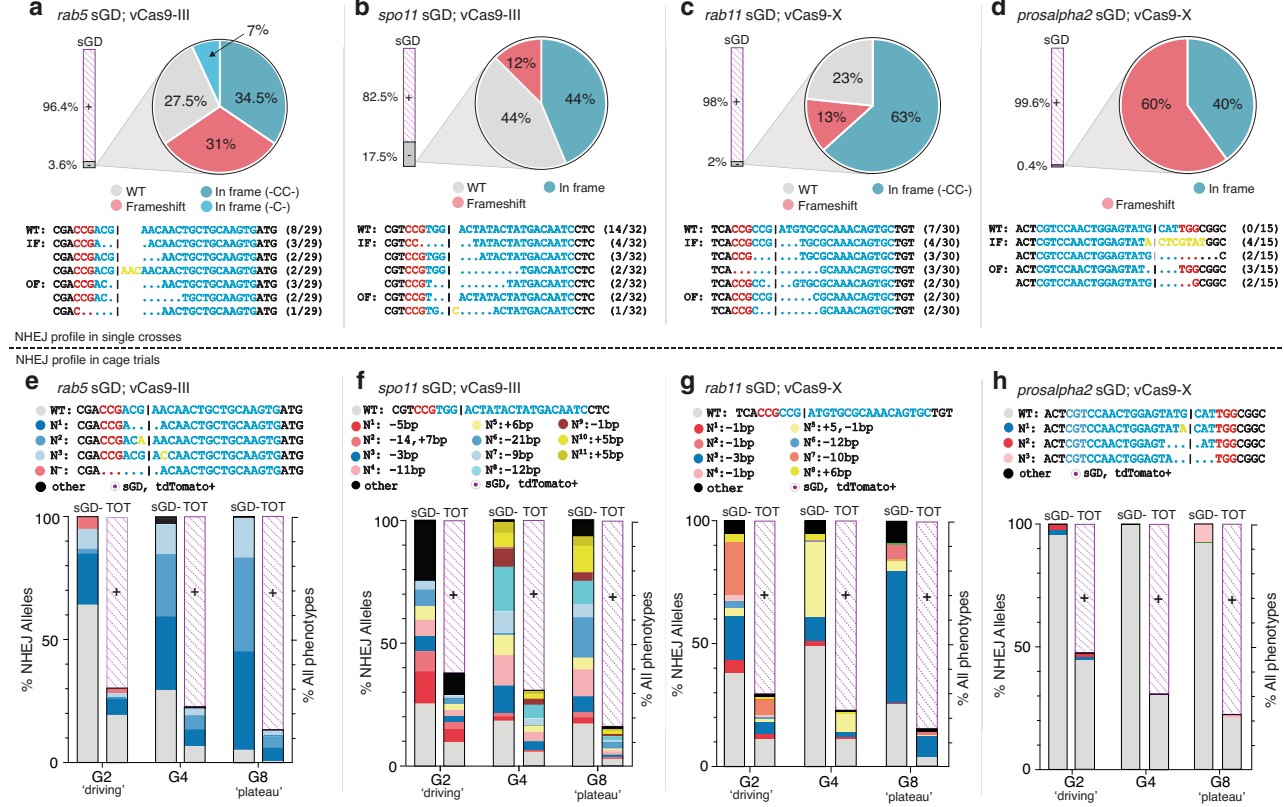

**Fig. 3 Profiles of NHEJ events in single-generation crosses versus multigenerational cages.** Production of NHEJ events in single crosses (**a–d**) and cage trials (**e–h**). For single-cross data, target regions were amplified from single non-fluorescent $F_2$ individuals generated in Fig. 2, sequenced through Sanger sequencing and analyzed. A bar depicts the % of sGD$^+$ (purple) and % of non-fluorescent (sGD$^-$, gray) flies for each tested locus. Genotypic data is depicted in pie charts representing the prevalence of specific indel mutations in sGD$^-$ individuals for **a** rab5, **b** spo11, **c** rab11, **d** prosalpha2 loci. Each section of the pie chart describes the kind of NHEJ that is formed and its percentage among the total tested sGD$^-$ (NHEJ/WT) heterozygotes. The specific sequence of prominent NHEJ events, along with its corresponding prevalence, is reported under each pie chart. gRNA sequence of each sGD is depicted in blue with its PAM sequence shown in red. To generate the NHEJ cage trial data (**e–h**), non-fluorescent sGD$^-$ individuals were pooled at every generation and used to amplify their target site region, which was deep sequenced to assess formation of NHEJ alleles. WT sequences (gray), in frame deletions (blue), frameshift deletions (red) and insertions (yellow) are shown in bars at each generation to represent the distribution of alleles in the sGD$^-$ population (left) and among the total population (sGD$^+$ and sGD$^-$, right). Purple diagonally-dotted bars show the sGD$^+$ population percentage.

(Fig. 3f), similar patterns of NHEJ alleles persisted throughout the sequenced generations. Thus, the spo11 locus may be significantly less constrained by selection than rab5 tolerating a greater array of alleles with reduced activity (in or out-of-frame). Alternatively, particular rab5 alleles may become fixed in a biased fashion by outcompeting other NHEJ events that carry greater fitness costs.

Another salient trend in the rab5 and spo11 cages was the gradual progressive reduction in the prevalence of the Cas9 transgene in many cage replicates. In the rab5 sGD experiments, Cas9 prevalence decreased with variable kinetics among cage replicates, whereas similar slopes of Cas9 decrease were observed in all of the spo11 sGD cage trials. It is notable that following the disappearance of Cas9, whenever it occurred, the prevalence of the rab5 and spo11 sGDs remained stable in subsequent generations (Fig. 4a, b), indicating that the genetic recoded alleles carried by these elements bore no substantial fitness costs relative to other alleles present in the cages (either functional NHEJ or WT alleles of the target locus). Because we hypothesized that lethal (rab5) or sterile (spo11) mosaicism may contribute to the drive dynamics, we determined rates of egg-laying, hatching and egg-to-adult ratios for the different sGD±Cas9. All percentages were normalized to hatchability and adults observed on WT crosses. We observed reduced hatchability and development to adulthood for trans-heterozygote rab5 sGD; vCas9-III (84% and 73.7%, respectively) and spo11 sGD; vCas9-III (91.4%

and 81.4%) crosses relative to both sGD/+ (95.5% and 92.2%), vCas9-III (100.5% and 95.3%) crosses (Supplementary Fig. 5). The higher hatching rates of the spo11 sGD drive compared to the rab5 sGD may reflect the differential effects of lethal versus sterile mosaicism (see discussion).

The drive dynamics for rab11 sGD cages differed from those of the rab5 and spo11 sGDs in that, accompanying an increase of the sGD to a plateau level of 80% prevalence from $G_8$ through $G_{20}$, vCas9-X remained constant in frequency (Fig. 4c). The higher proportion of in frame NHEJ alleles observed in single crosses (Fig. 3c) for the rab11 sGD compared to the rab5 or spo11 sGDs may account for its reduced level of maximal introduction, even with stable maintained inheritance of the vCas9-X transgene. Lending further support to this interpretation, deep-sequencing of non-fluorescent flies at different generations (Fig. 3g) revealed that a variety of mutations in rab11 were generated during the drive process, only a small fraction of which fixed in the population by $G_8$. At that point, nearly all remaining non-fluorescent flies contained a single-codon deleted allele which presumably encoded a functional protein variant lacking a single amino acid (ΔAsp).

**The prosalpha2 sGD displays differential drive behavior depending on the Cas9 source.** As described above, the rab5 and

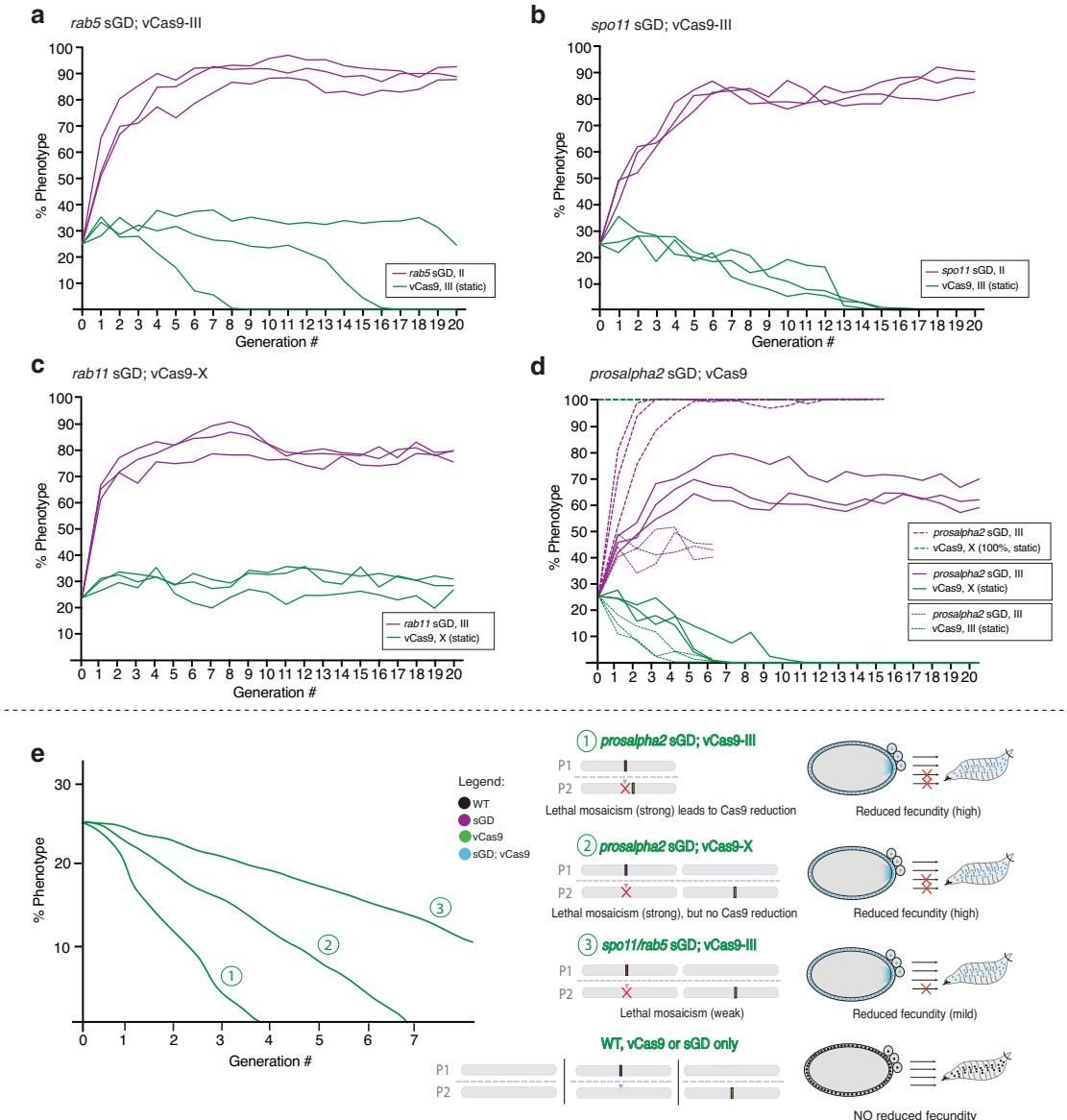

**Fig. 4 sGD driving experiments in cage trials.** Virgin sGD/Cas9 trans-heterozygotes and WT flies were seeded at a 1:3 ratio. Each generation, flies in a cage were randomly split in half. One half was scored for eye fluorescence ($G_n$) while the other was used to seed fresh cages ($G_{n+1}$). Purple traces indicate sGD+ progeny, green traces indicate Cas9+ progeny. Experiments were done in triplicate and each line represents a separate cage. **a** *rab5* sGD; vCas9-III. The sGD prevalence increases exponentially in the cage (84±6%) up to $G_4$ and then plateaus slowly. sGD highest percentage occurs at $G_{10}$ (92 ± 4%). Cas9 decreases in two of the three cages from 25% to 0% by $G_8$ and $G_{15}$, respectively. **b** *spo11* sGD; vCas9-III. All three replicates reach their highest prevalence in the cage (85 ± 2%) by $G_{6-7}$ and then plateau. Cas9 decreases linearly from 25% to 0% by $G_{15}$ in all three replicates. **c** *rab11* sGD; vCas9-X. sGD proportions in the cage slowly increase linearly (83±6%) up to $G_8$. Cas9 remains steady at seeding levels (28 ± 3%), suggesting continuous Mendelian transmission. **d** *prosalpha2* sGD drive dynamics depend on the location of the static Cas9, as well as seeding ratios. Bold lines reflect sGD; vCas9-X, thin dashes show cage trials using vCas9-III and thicker dashes depict drive in a vCas9-X-saturated population. *Prosalpha2* sGD; vCas9 combinations produce different driving fates and outcomes, providing a flexible tool for deployment. **e** Hypothesis on cage and drive behavior of the different sGDs and vCas9 reduction over time. Raw phenotypic scoring is provided as "Supplementary Data 2".

*spo11* sGD elements exhibited similar drive trajectories in cage experiments, consistent with their performance in single-generation crosses (considering also that the *spo11* sGD is expected to benefit from sterile mosaicism a generation later than *rab5* sGD, which profits from lethal mosaicism). In contrast, *prosalpha2* sGD; vCas9-III cages displayed a curtailed trajectory in which the drive leveled off much sooner than for the other drives (Fig. 4d, thin dashed lines). This population-level behavior was surprising as it contrasted markedly with the efficient transmission of this sGD through both males and females in single-generation crosses, where it displayed the greatest drive

potential, using the vCas9-III source (>99%). Yet, with the same Cas9 strain in cages, this sGD achieved only a modest level of introduction (Fig. 4d, thin dashed lines). Another notable feature of all three cage replicates was a very rapid decline of the Cas9 transgene (lost in all cages by $G_6$), in line with the sub-Mendelian (<50%) transmission of the genetically-linked vCas9-III transgene observed in single crosses (Fig. 2d). We further analyzed the drive performance of the *prosalpha2* sGD using an X-linked source of Cas9 (vCas9-X), unlinked to the sGD transgene. vCas9-X also declined over time when combined with the *prosalpha2* sGD, although did so more gradually than with its vCas9-III

counterpart (Fig. 4d, bold lines). Accordingly, the *prosalpha2* sGD reached higher levels of introduction with the unlinked vCas9-X (60-70%, Fig. 4d, bold lines) than it did with vCas9-III, yet still fell short of that achieved by the *rab5* sGD or *spo11* sGD driven by the action of vCas9-III. However, unlike the *rab5* and *spo11* sGD drives, in which functional NHEJ alleles were generated and selected, very few such NHEJ alleles accompanied the partial spread of the *prosalpha2* sGD (Fig. 3h). We observed the transient generation of a small number of varied mutations at $G_2$ and $G_4$. By $G_8$, nearly all remaining non-drive alleles had the WT sequence that should remain susceptible to conversion. The decline in vCas9-X in these experiments contrasts with its stable maintenance in *rab11* sGD in cages, and with its Mendelian inheritance in single-generation *prosalpha2* sGD crosses. Hatching rates (79.8%) from *prosalpha2* sGD; vCas9-X mothers revealed the greatest reduction among all sGDs relative to heterozygous *prosalpha2* sGD/+ (99.2%) and vCas9-X/+ (99.4%) controls (Supplementary Fig. 5). Similarly, decrements in embryonic survival (65.3%) and adult viability (80% of the hatched larvae) were observed. This substantial reduction in fecundity most likely contributes to the observed cage dynamics, particularly when acting over several generations in populations mating randomly.

Since the *prosalpha2* sGD displayed the greatest transmission in single-generation crosses (>99%, Fig. 2d), we wondered whether we could increase its level of introduction in population cage experiments given that nearly all non-drive alleles that remained following elimination of the Cas9 source were WT. We tested this possibility by seeding *prosalpha2* sGD; vCas9-X drive competent individuals into a population of pure vCas9-X homozygous flies. This scenario forces maintenance of Cas9 in the population, approximating a full drive configuration in this respect. We seeded cages at the same 25% trans-heterozygous sGD; vCas9 rate as in the other cage experiments, but replaced the WT population with vCas9-X homozygous flies, thereby increasing initial Cas9 prevalence to 100%. In contrast to the self-attenuated drive manifested in the previous examples, the sGD now rapidly achieved complete fixation in 3 generations in 2 of the 3 cages, and in 5 generations in the third cage (Fig. 4d, thick dashed lines). No NHEJ events were recovered from the *prosalpha2* sGD; vCas9-X cages (100% vCas9) analyzed for these events. This absence of mutant alleles is consistent with the highly-efficient drive trajectory of the *prosalpha2* sGD in a Cas9 background and with the large fraction of unaltered target alleles remaining in cages when this sGD was seeded into WT populations. The divergent drive outcomes under these differing scenarios suggests a hypothesis for behavior of the *prosalpha2* sGD (Fig. 4e) that may also pertain in a less potent mode to other sGDs, as discussed below.

**Modeling captures varied cage drive dynamics and reveals a gradient of allelic fitness costs.** We generated four mathematical models to account for the results observed in single generation crosses and cage trials of sGDs inserted into: 1—an autosomal locus required for fecundity (*spo11*, vCas9-III), 2—an autosomal sGD locus essential for zygote viability (*rab5*, vCas9-III), 3—linked autosomal sGD and Cas9 elements with the sGD inserted in a gene essential for zygote viability (*prosalpha2*, vCas9-III), and 4—an X-linked Cas9 source combined with sGDs required for zygote viability (*rab11* and *prosalpha2*, vCas9-X). Drive efficacy was refined and fitness cost parameters were estimated from respective cage trial data using predictive mathematical models and a likelihood-based Markov chain Monte Carlo (MCMC) algorithm, with initial parameter estimates taken from single-pair mating data (Supplementary Figs. 6–11). Model-estimated rates

for cleavage and accurate HDR events were consistent with frequencies observed in single-pair crosses shown in Fig. 2, except for *rab11* (Supplementary Tables 8–13). Modeled HDR frequencies for *rab11* females were significantly lower (~8%) than those observed in single-pair crosses, and slightly lower in males (~2%). Also, we note that the initial drive ascent in the modeling is somewhat less than the observed experimentally for the several of the sGDs, which may reflect an initial mating bias with females transmitting the gene drive more efficiently than males as seen in single-generation crosses. Consistent with such a density-dependent phenotype mechanism, the modeling more closely resembles the early dynamics of the *prosalpha2* sGD experimental cages, which display little sex-biased transmission in single crosses. The parameters describing the proportion of generated NHEJ alleles that are functional vs. non-functional generally had a broad credible interval (CrI), e.g., for the X-linked Cas9 - *rab11* split-drive design, the proportion of functional NHEJ alleles that were retained under selection was estimated to be 95.7% (95% CrI: 69.8–99.9%) in females and 99% (95% CrI: 75.1–99.9%) in males. Lack of resolution for this parameter stems from the high rate of accurate HDR and hence relatively low number of NHEJ resistant alleles. Estimated fitness costs demonstrate the potential for deliberate tuning through drive design and target gene selection. Different target genes showed various fitness costs, with *rab11* sGD estimated to have the lowest associated cost combined with Cas9 (0.0%, 95% CrI: 0.0–0.1%), followed by the *rab5* sGD (4.8%, 95% CrI: 1.0–5.1%) and *spo11* (11.3%, 95% CrI: 10.6–12.3%), and the *prosalpha2* sGD with the highest cost (15.8%, 95% CrI: 15.1–16.2%) (Supplementary Tables 8–13). Using two designs for *prosalpha2* (an autosomal split-drive with vCas9 located on the same chromosome (III) and a second using an X-linked vCas9), we found that targeting a chromosome *in cis* was costlier than targeting a chromosome *in trans* for this locus (cost of 27.5%, 95% CrI 26.0–28.3% *in cis* c.f. 15.8%, 95% CrI 15.0–16.2% *in trans*). Collectively, these findings suggest it may be possible to vary fitness costs by design through thoughtful pairing of a target gene and Cas9 source, with the caveat that fitness costs measured under laboratory conditions may differ from those experienced by organisms in the field.

Finally, employing model-estimated drive efficacy and fitness cost parameters (Supplementary Figs. 6–11), predictions from a stochastic implementation of the model were compared to experimental cage trial data (Fig. 5). The stochastic model captured the potential role of chance events such as mate choice (multinomial-distributed), egg production (Poisson), progeny genotype (multinomial), and the finite sampling of the next generation (multivariate hypergeometric). Stochastic model trajectories were consistent with the observed experimental data for each construct, both in terms of mean behavior and stochastic variation. This provides reassurance that: (1) the model is capturing dynamic features pertaining to each of the drive systems, (2) the variation observed in the laboratory cage trials can be explained by stochasticity given cage population sizes and fitness costs involved, and (3) plausible relative fitness costs associated with sGD$^+$ Cas9 combinations can be inferred.

## Discussion
In this study, we design and test a range of inherently confinable split drives targeting essential recessive genes. These sGD elements carry recoded target gene sequences that both mitigate the formation of cleavage-resistant alleles and lead to the loss of separately-encoded Cas9 transgenes in population experiments. Specifically, the elimination of cleavage-resistant alleles relies on LOF NHEJ events being rapidly eliminated after NHEJ alleles are created by dominantly acting maternal lethal-sterile

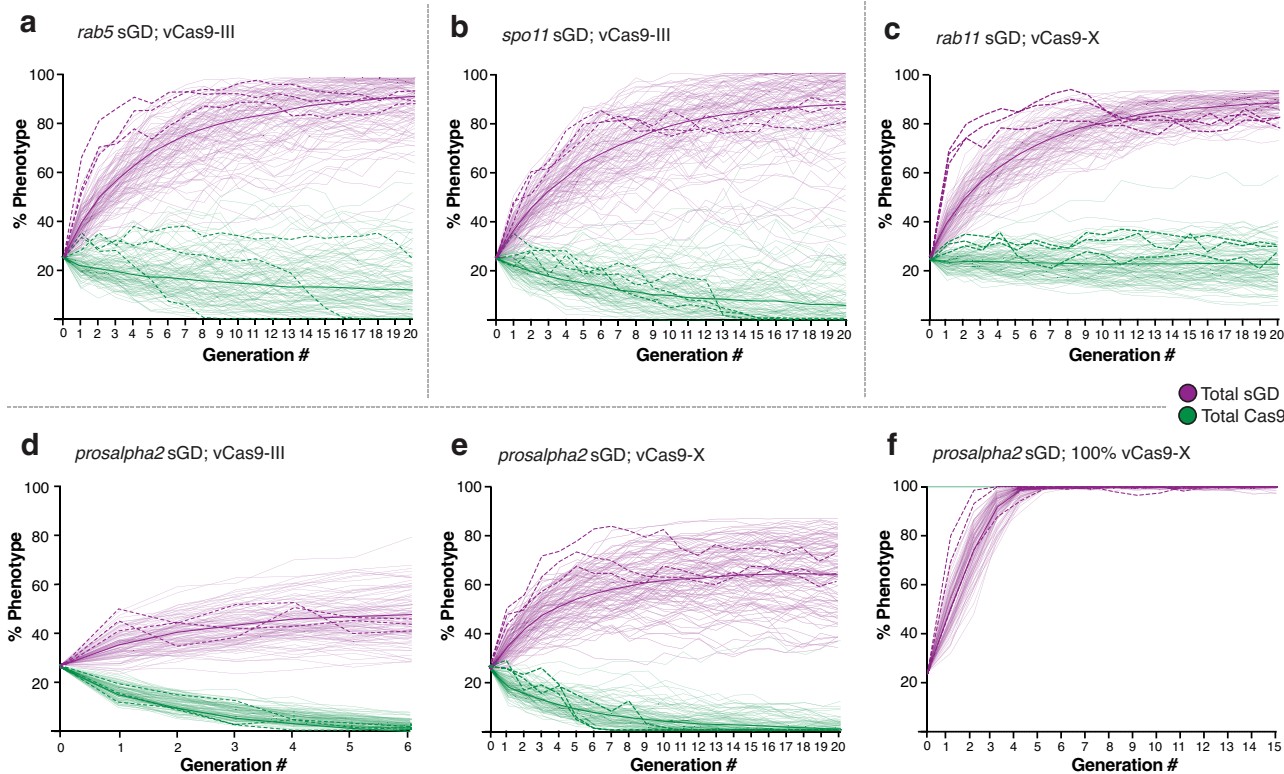

**Fig. 5 Mathematical model simulations recapitulate cage trial experimental data.** Four mathematical models were designed based on target gene biology and behavior: split drive in autosomes targeting viability (**a**), split drive in autosomes affecting fecundity (**b**), linked split drive (**d**) and X-linked Cas9 split drive targeting viability (**c**, **e**, **f**). Simulations were run using estimated and fitted parameter values and 100 stochastic model realizations depicted in thin purple (sGD) and green (Cas9) lines. Thicker lines show the mean of those 100 simulations. For comparison, dashed curves represent the collected experimental data.

mosaicism[2,22], followed by gradual culling by standard Mendelian homozygosis and/or moderate haplo-insufficient phenotypes. The rapid loss of Cas9 transgenes appears to depend on two putative mechanisms: (1) reduced viability or fecundity of individuals carrying both a Cas9 source and certain sGDs due to partial haplo-insufficiency of the locus and/or off-target Cas9/gRNA affects, and (2) damage to the Cas9/gRNA-targeted chromosome (if linked genetically to the Cas9 source). We hypothesize that these distinct mechanisms may reduce the overall fitness of sGD[+]; Cas9[+] individuals.

All sGDs tested in single-generation crosses displayed significant super-Mendelian inheritance ranging up to ~100%. One clear trend from these experiments was that all sGDs (except *prosalpha2*) exhibited significantly higher inheritance rates (~15%) when transmitted by $F_1$ master females than by $F_1$ master males. Since many components of the HDR DNA repair pathway are shared between repair of damage-induced DSB and meiotic recombination[35], these shared features may underlie both the peculiar lack of male recombination in *Drosophila* and the reduced rates of HDR-mediated gene conversion resulting in the divergent sex-specific sGD drive frequencies we observed. This phenomenon was also observed in other studies using a full gene drive[12], as well as in a trans-complementing framework[27]. Why this sex difference was not also observed for the *prosalpha2* sGD will require further analysis, but may be related to a combination of highly-efficient copying and particularly strong forms of lethal mosaicism and/or haplo-insufficiency that eliminate nearly all LOF alleles (NHEJs or damaged chromosomes) when generated. It is noteworthy that when the *prosalpha2* sGD drive was placed *in trans* to a vCas9-III source located on the homologous 3rd chromosome at a site tightly linked to the *prosalpha2*

locus, the vCas9 source was inherited at a sub-Mendelian rate in both males and females, suggesting that the target chromosome may have been damaged by an imperfect repair process leading to its loss during or shortly after transmission. Additionally, the two III-chromosome sources of Cas9 may be expressed either at higher levels or in temporal patterns that promote greater overall lethal/mosaic activity.

The different sGD; Cas9 combinations displayed varying levels of drive in population cages. The *rab5* and *spo11* sGDs demonstrated intermediate levels of drive resulting in introduction of the sGD into 80-95% of the population accompanied by a gradual loss of the vCas9-III transgene. This Cas9 elimination mechanism (summarized above regarding the *prosalpha2* sGD) is inferred to operate at varying levels of intensity for the *prosalpha2*, *rab5*, and *spo11* sGDs since we only observed multigenerational loss of Cas9 or reductions in egg-laying and hatching when the sGD element and Cas9 were carried together (when sGD conversion occurs), but not for each of the separate elements. These results demonstrate that sGDs targeting genes essential for viability or fertility can effectively spread in a super-Mendelian fashion, provided that a recoded version of the gene is carried by the propagating sGD element and that the gRNA site is chosen to target functionally-critical elements of the gene so as to generate as few functional NHEJ alleles as possible. Such loci can be used as docking sites for a split form of drive as shown here, as well as a full gene-drive system or in other strategies such as integral gene drives[36], Medea-like toxin-antitoxin systems[37–40] or daisy-chain drives[41]. We note that our system was designed to be readily converted to a full-drive configuration using "homology assisted CRISPR knock-in"[29] to create a self-spreading full-drive element (gRNA and Cas9 would spread together), which is similar in outcome to the

RMCE strategy in which a Cas9 transgene was mobilized into pre-existing docking sites[3]. Indeed, we have obtained preliminary proof-of-principle data that such conversion to autonomous drives can be achieved (for the *spo11* sGD), and a comparison between such full and split-drive versions of these recoded drives will be assessed in future studies.

The behavior of the *prosalpha2* sGD in population cages did not conform to simple expectations based on its performance in single-generation experiments. We observed different behaviors of this sGD depending on the source of Cas9. When seeded with an autosomal vCas9-III at a 25% initial frequency, little sGD drive was observed and the Cas9 transgene was rapidly eliminated from the population (within 3 generations). This precipitous decrease in Cas9 levels may reflect the damage incurred by the receiver chromosome, evidence for which was observed in single-generation crosses (Fig. 2d), as well as by fitness costs associated with carrying both elements. As noted above, the vCas9-III source also declined to varying degrees in different cage replicates when coupled with the other sGDs. Combining the *prosalpha2* sGD with an unlinked Cas9 source (vCas9-X) resulted in increased drive, but fell short of the levels attained by the *rab5* and *spo11* sGDs, most likely due to the fitness costs of carrying vCas9-X and the sGD, which again resulted in speedy elimination of the vCas9-X element from the population. It is noteworthy that the vCas9-X source remained stable in the presence of the *rab11* sGD, indicating that this Cas9 source carries little if any fitness burden on its own or coupled with this particular sGD. Importantly, in *prosalpha2* sGD cages employing the less aggressive vCas9-X source, most of the remaining non-fluorescent fly population carried WT alleles. This retention of WT alleles suggests that further spread of the sGD transgene should occur upon a second introduction of trans-heterozygote or Cas9-only flies in the population, providing a serially-dosable dynamic drive system. Indeed, this prediction was born out when the *prosalpha2* sGD was seeded into a homozygous vCas9-X population, leading to a rapid and complete introduction of the drive (~3 generations with a 12.5% allelic seeding frequency).

We offer the following hypothesis to account for *prosalpha2* sGD cage results (Fig. 4e) and their relevance to performance of the other sGDs. According to this model, the *prosalpha2* sGD incurs two types of drive-dependent fitness costs in different scenarios. When crossed to vCas9-III, it both damages the target chromosome (as evident in the sub-Mendelian inheritance of the Cas9 target chromosome in single-generation crosses) and also induces a significant heterozygous fitness cost. The latter fitness cost, estimated by modeling, may be associated with efficient and penetrant lethal mosaicism leading to moderate haplo-insufficient and/or off-target phenotypes and consequent potent selection against individuals carrying both Cas9 and the *prosalpha2* sGD. These two processes act in concert to rapidly reduce Cas9 prevalence. When the *prosalpha2* sGD is combined with the unlinked and presumed less active vCas9-X source, it does not inflict appreciable damage to the Cas9 chromosome (which displays Mendelian transmission), but still generates a sGD/Cas9-dependent fitness lethal mosaic cost resulting in a more gradual loss of the Cas9 transgene. This combined sGD/Cas9 effect is also displayed albeit in milder forms (estimated again by modeling) by the *rab5* and *spo11* sGDs.

In aggregate, this study offers insights into important design features of the sGD system that can be exploited to achieve specific levels of spread and confinement of these systems when applied into native populations. Thus, modest introduction of the sGD followed by rapid removal of the endonuclease can be achieved by placing the *prosalpha2* sGD across from a strongly-linked Cas9 source. Intermediate levels of drive result from moderating the intensity of lethal mosaicism or by employing

sterile mosaicism to delay fitness penalties by one generation. Full population introduction can be attained by increasing the proportion of a moderately active Cas9 source to the *prosalpha2* sGD. Such tunable systems could serve as updating platforms to sustain allelic drives[22] that bias inheritance of beneficial traits such as susceptibility to insecticides, or to disseminate cargo with desired features such as new rounds of anti-pathogenic molecules to supplement those carried by an original modification drive[42,43].

## Methods

**Plasmid construction.** All plasmids were cloned using standard recombinant DNA techniques. Recoded cDNA fragments were designed by using non sub-optimal alternative codons from cut site to stop codon and thus maintaining the exact amino acid sequence of each target gene upon insertion. In all instances, codon usage was kept as similar as possible to that of the endogenous sequence. All fragments were synthesized as gBlocks™ (Integrated DNA Technologies) and cloned into the desired vector. Plasmid and genomic DNA sequences were amplified using Q5 Hotstart Master Mix (New England Biolabs, Cat. #M0494S) and Gibson assembled with NEBuilder HiFi DNA Assembly Master Mix (New England Biolabs, Cat. # E2621). Resulting plasmids were transformed into NEB 5-alpha chemically-competent *E. coli* (New England Biolabs, Cat. # C2987), isolated and sequenced. Primer sequences used for the creation of the different plasmids can be found in Supplementary Table 1.

**Microinjection of constructs.** Plasmids were purified using the PureLink Fast Low-endotoxin Maxi Plasmid Purification kit (ThermoFisher Scientific, Cat. #A35895). All constructs were fully re-sequenced prior to injection. Embryo injections were carried out at Rainbow Transgenic Flies, Inc. (http://www.rainbowgene.com). Each gRNA construct was injected into a vasa-Cas9 expressing line in the 3rd chromosome (Bloomington #51324).

Injected embryos were received as $G_0$ larvae, were allowed to emerge and 3–4 females were intercrossed to 3–4 males in different tubes. $G_1$ progeny were screened for the eye-tdTomato positive marker that indicates transgene insertion. All transgenic flies that displayed the red marker were then balanced using Sco/CyO (for genes located on the II chromosome) or TM3/TM6 (III chromosome) and kept on a $w^{1118}$ background. Homozygous stocks were kept in absence of any balancer alleles. Correct insertions in homozygous transgenic stocks were confirmed through Sanger sequencing using primers found in Supplementary Table 1.

**Fly genetics and crosses.** Fly stocks were kept and reared on regular cornmeal medium under standard conditions at 20–22 °C with a 12-hour day–night cycle. sGD housekeeping and crosses were performed in glass vials in an ACL1 fly room, freezing the flies for 48 h prior to their discard. To assess each locus' copying efficiency, we genetically crossed each sGD construct to different Cas9 lines. Since our genes of interest are autosomal, individual trans-heterozygote $F_1$ males or virgin females were collected for each $G_0$ cross and crossed to a wildtype fly of the opposite gender. Single one-on-one crosses were grown at 25 °C. Inheritance of both gRNA (>50%) and Cas9 (~50%) were calculated using the resulting $F_2$ progeny by scoring the phenotypic markers associated to each transgenic cassette.

**Multigenerational cage trials.** All population cage experiments were conducted at 25 °C with a 12-hour day-night cycle using 250 ml bottles containing standard cornmeal medium. Crosses between flies carrying the gRNA and flies carrying vCas9 (X or III) were carried out to obtain $F_1$ trans-heterozygotes used to seed the initial generation. Wildtype or trans-heterozygote males and virgin females were collected and separately matured for 3–5 days. Cages for all loci were seeded at a phenotypic frequency of 25% gRNA/Cas9 trans-heterozygotes (15 males, 15 females) to 75% $w^{1118}$ (45 males, 45 females). In each generation, flies were allowed to mate and lay eggs for ~72 h, when parents were removed from the cage ($G_n$), and kept for 10 days. Subsequent progeny ($G_{n+1}$) were randomly separated into two pools and scored; one was collected for sequencing analyses while the other was used to seed the following generation. If the two pools differed much phenotypically, frequencies were averaged in order to reduce variability and stochastic extremes. This process of sampling and passage was continued for 10–20 generations.

**Molecular analysis of resistant alleles.** To extract fly genomic DNA for single fly resistant allele sequence analysis, single flies were squashed in lysis solution (10 mM Tris-Cl pH 8.2, 1 mM EDTA, 25 mM NaCl and 0.2 mg/ml proteinase K), incubated at 37 °C for 30 min and deactivated at 95 °C for 2 min, as described previously[44]. After extraction, each sample was diluted 1:3 (sample:$H_2O$) and stored at −20 °C if needed. 1–2 μl of each diluted DNA extraction was used as template for a 25 μl PCR reaction that covered the flanking regions of the gRNA cut site, which were used to sequence the alleles. Sanger sequencing in individual

non-fluorescent flies was performed at Genewiz, Inc. in San Diego, CA to obtain the single-cross NHEJ data. NHEJ allele sequences were obtained from Sanger chromatographs by isolating the WT sequence first and then annotating the remaining allelic sequence. For cage trials, 20 non-fluorescent flies were pooled together and DNA was extracted at each sampling generation (2, 4 and 8). Target sequences were amplified using specific gene primers that also contained adapter sequences. Non-fragmented amplicons were sequenced using Illumina-based technology (2x250bp reads, Amplicon-EZ, Genewiz), with final data being delivered as FASTQ reads and aligned to a reference sequence for each gene of interest to detect indel formation. For all datasets, an average of 72,594 ± 8712 (mean ± SD) paired-end amplicon reads were obtained per sample.

Primer sequences used for either single fly or cage trial deep sequencing analyses can be found in Supplementary Table 1.

**Viability assays**. For embryo viability counts, 2 to 3-day old virgin female flies were mated to wild-type males for 24 h and 15 females (in triplicate) were placed in egg-collection chambers to lay eggs during a 18 h period, then the laying plate was removed and eggs counted. All embryos were counted and kept on an agar surface at 20 °C for 48 h and hatchability of those eggs was calculated at that point by counting the unhatched embryos. In all plates, females laid between 211 and 332 eggs. Each experiment was carried out in triplicate, and the results presented are averages from these three experiments. For adult fly counts, the larvae obtained in each embryo count assay replicate were transferred from egg collection plates to 250 ml bottles containing modified cornmeal medium. All adult flies that emerged from these bottles were counted and the results of the three replicates for each experiment averaged together. Egg-to-larvae and egg-to-adult ratios were calculated by normalizing the experimental values to the survival observed in parallel experiments carried out with WT flies.

**Mathematical modeling**. Model fitting was carried out for all four constructs and six distinct cage trials, using a discrete-generation adaptation of the Mosquito Gene Drive Explorer (MGDrivE)[45]. A likelihood-based Markov chain Monte Carlo (MCMC) procedure was used to estimate gene drive efficacy and genotype-specific fitness costs, providing Maximum a Posteriori (M.A.P.) estimates and 95% credible intervals for each parameter of each distinct cage trial[46]. Mendelian inheritance was assumed except under co-occurrence of the Cas9 and gRNA constructs, when the split-drive design allowed active cleavage of the target chromosome and the possibility of super-Mendelian inheritance. When cleavage occurred, a fraction of the cut alleles could be properly repaired via HDR, and the remaining cut alleles underwent NHEJ repair, generating *in* or *out-of-frame* resistant alleles. The effects of shadow-drive, in which Cas9 protein is deposited in the embryo of a female individual who does not carry the Cas9 allele, but whose own mother does, were also accommodated. Fitness costs were implemented as fractional reductions in male and female fecundity (*spo11*) or male-mating competitiveness and egg viability reductions (*rab5*, *rab11*, and *prosalpha2*), again only under active cleavage conditions with both the Cas9 and gRNA alleles present. In the model, it was assumed that any individual with two *out-of-frame* alleles was completely infertile (*spo11*) or did not mature past the egg stage (*rab5*, *rab11*, *prosalpha2*). More details on the model implementation and likelihood function used in the model fitting can be found in the Supplementary Information file (as a Supplementary Method). The results of each fit, using the estimated parameters, are plotted in Supplementary Figs. 6–11, along with corresponding experimental cage trial data. Parameter descriptions and estimates for each cage trial are provided in Supplementary Tables 2–13.

Simulated model trajectories for Fig. 5 were generated using a stochastic implementation of the discrete-generation model. At each generation, adult females mate with males, thereby obtaining a composite mated genotype (their own, and that of their mate) with mate choice following a multinomial distribution determined by adult male genotype frequencies, modified by mating efficacy. Egg production by mated adult females then follows a Poisson distribution, proportional to the genotype-specific lifetime fecundity of the adult female. Offspring genotype follows a multinomial distribution informed by the composite mated female genotype and the inheritance pattern of the gene drive system. Sex distribution of offspring follows a binomial distribution, assuming equal probability for each sex. Female and male adults from each generation are then sampled equally to seed the next generation, with sample size proportional to the average size of the cage trials at that generation, following a multivariate hypergeometric distribution. All simulations were performed and analyzed in R[47].

**Figure generation and statistical analysis**. Graphs were generated using Prism 8 (GraphPad Software Inc., San Diego, CA) and potentially modified using Adobe Illustrator CS6 (Adobe Inc., San Jose, CA) to visually fit the rest of the non-data figures featured in the paper.

**Safety measures**. All sGD crosses were performed in glass vials in an ACL1 facility, in accordance with the Institutional Biosafety Committee-approved protocol from the University of California San Diego. All vials were frozen for 48 h prior to autoclaving and discarding the flies.

**Reporting summary**. Further information on research design is available in the Nature Research Reporting Summary linked to this article.

## Data availability
The sequence of all sGD constructs generated in this paper has been deposited into the GenBank database with the following accession codes: MW540521 (*prosalpha2*), MW540522 (*rab5*), MW540523 (*spo11*), MW540524 (*rab11*). Source data is provided in this paper as a Supplementary Data files (1–3). The rest of the data is available from the authors upon request.

## Code availability
Modeling information and parameters can be found as a Supplementary Method of this study in the Supplementary Information file. A version of MGDrivE was used for simulation modeling and is freely available from the MGDrive GitHub repository (https://marshalllab.github.io/MGDrivE/). Specific code can be obtained from the authors upon request.

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

## Acknowledgements

We would like to thank members of the Bier, Akbari and Gantz laboratories for continuous comments and discussions on the paper. We thank Ting Yang for her contribution on generating reagents, Nikolay P Kandul for insights and providing nCas9 fly lines, Carissa Klanseck for helping with cage trial counts and Xiang-Ru (Shannon) Xu for her help and comments on figures. O.S.A., A.B.B., I.S., J.B.B. and J.M.M. were supported in part by funding from a DARPA Safe Genes Program Grant (HR0011- 17-2-0047) and O.S.A., A.B.B. and I.S. by NIH grants (R21RAI149161A, R01AI151004, DP2AI152071) awarded to O.S.A. These studies were supported by NIH grant R01GM117321, a Paul G. Allen Frontiers Group Distinguished Investigator Award to Ethan Bier (EB) and a gift from the Tata Trusts in India to TIGS-UCSD.

## Author contributions

G.T., A.B.B., O.S.A. and E.B. conceptualized the study. G.T. and A.B.B. contributed to the design of the experiments. G.T., A.B.B. and I.S. cloned the constructs, isolated the transgenic lines and gathered preliminary data. G.T. performed the experiments and analyzed the data. J.B.B. and J.M.M. performed mathematical model fitting and simulations. G.T. designed all artwork and figures for the paper. All authors contributed to the writing and approved the final manuscript.

## Competing interests

E.B. has equity interests in Synbal and Agragene, companies that may potentially benefit from the research results and also serve on the company's Scientific Advisory Board and Board of Directors. O.S.A. also has equity interests in Agragene. The terms of this arrangement have been reviewed and approved by the University of California, San Diego in accordance with its conflict of interest policies. The remaining authors declare no competing interests.
