## [Peer Review File · Nature Communications]

Reviewers' Comments:

Reviewer #1:

Remarks to the Author:

Summary:

The authors present data describing the performance of several Cas9-based split Gene Drives (sGDs) in the model organism *D. melanogaster*. The design is novel in that they target genes essential for viability or fertility and simultaneously introduce recoded versions of the targeted sequences with the homing drive. This creates a system in which undesired NHEJ-repaired alleles are cleared via a dominantly-acting maternal inheritance of active Cas9 that creates lethal/sterile mosaics in the next generation.

The split-drive architecture allowed for a combinatorial generation of trans-heterozygote 'master strains' to empirically test many different design variations. The authors do a good job of highlighting the nuanced ways in which genetic context impact the performance of specific drives throughout the results and discussion. These genetic context effects were not examined mechanistically in this paper (i.e. with careful measurements of Cas9 expression from different chromosomal loci) but this does not detract from the overall message of the paper.

This study is well-designed, well-communicated, and complete. The authors provide an end-to-end report of their work that includes system design, construction and testing, population-level experiments, and modeling. Their conclusions are supported by ample experimental data. I think this paper will be well-received by the field.

There are suggested improvements listed beneath my signature. I recommend this paper for publication in Nature Communications with minor revisions (i.e. no additional experiments). **I welcome questions or concerns about the content of this review by the authors of the manuscript or editors at Nature Biotechnology.** If it is against editorial policy to have non-anonymous reviews, please remove my name below (although I will note that I include my contact information in all peer-reviews and it has only led to neutral or positive outcomes so far).

Substantive points to consider during revision:

1. The readability of the paper could be improved to make this study more accessible. Several specific examples are given in the minor points section below. Two common themes would be to (i) be more concise with the description of the results. The current word count is probably ~60% over the Nature Communications suggested limit of 5,000 words. And (ii) the figure legends often contain much more information than is needed to understand the figure. I suggest not duplicating statements about the results both in the main text and in figure legends.
2. The stochastic implementation of the mathematical model (i.e. Figure 5) routinely underestimates the drive efficiency for the first few generations of the simulations. It is notable that 3 of 3 experimental results appear as outliers to the 100 simulations in panels A and C, and 3 of 4 as outliers in panel F. While the experimental results in panel B do not reach the status of being outliers, the same trend seems to exist wherein the model is underestimating early drive efficiency. The authors' description of their modeling approach is sound, but the paper would be improved if the authors provided more text to explain this observation.

3. As a primary driver for this study seems to be the impact this system has on non-functional NHEJ alleles, it would be good for the authors to revisit the issue of overcoming genetic resistance in the context of real-world biocontrol scenarios. How low will resistant rates need to be for a field application? How does the rate of resistant alleles compare in this system versus other published systems? Are aspects of this system expected to work synergistically with other approaches (e.g. multiplexed guides)? The authors do this in part in the last paragraph, but it is difficult to glean the strengths and weaknesses of this drive design versus others.
4. Provision of a recoded version of the 3' region of the target gene CDS is an important aspect of this paper, yet no detail is given to the design of this recoded CDS. More details should be included, and possibly sequence comparisons of the wildtype CDS and recoded CDS in the SI. For example, is just the sgRNA target site recoded, or the whole fragment? How were variant codons assigned and what is the expected impact on translational elongation?

Minor points:

1. Line 69: you have two spaces after the word 'resistant'
2. Line 84: have -> has
3. Line 243: "Carrying both the Cas9 transgene and Cas9 source". I am not sure what is meant by this sentence. What is a Cas9 source, if not the cas9 transgene?
4. Line 300: delete the word 'unaltered'. It's inclusion indicates that there is such a thing as an 'altered WT allele', which would no longer be WT.
5. Line 401: Sentence needs to be revised somehow. Possibly to: "...given the remaining WT alleles present..."
6. Line 437: "through"
7. Line 655: correct in-text citation to a number corresponding with the list of references.
8. Line 665: Add a space between "18 h"
9. Figure 1b: the graphical difference between the 'different chromosome' and 'same chromosome' is quite subtle. I suggest adding a semicolon between the two chromosomes at left, which is compatible with typical genotype representations, would make the difference between the left and right drawings more obvious.
10. Figure 2: be consistent with use of significant figures. In the Mean Inheritance row (top) some experiments are reported with two significant figures, and others with three.
11. Figure 2 Legend: All text after "Super-Mendelian..." is a description of the results, not of the figure, and that text is better moved to the results section of the main text (Although it is probably already stated there, so can just be deleted).
12. Figure 3: Panel labels A,B,C,D are disproportionately large.
13. Figure 3 Legend: The sentence beginning "Each section.." is redundant to information earlier in the legend and can be removed.
14. Figure 4E: This part of the figure is difficult to interpret. The inset at top left is too small to see well, and the yellow dots in the equivalent figures at right are difficult to see. It is unclear why the word 'and' in the top example and the word 'but' in the bottom two examples. In general, I like the idea behind this panel (i.e. providing graphical explanation behind the different rates of loss of the Cas9 cassette), but unfortunately the execution of this panel is still a bit confusing.

15. Figure 4 Legend: There is a lot of text in the legend that is describing the results and would be better placed in the main text results section instead of the figure legend.
16. Figure 1,2,4,5: Use of red-green as the primary differentiation between data sets is not ideal for audience with red-green color blindness (8% of males of northern European descent).

Reviewer #2:

Remarks to the Author:

This manuscript looks at gene drives - selfish genetic elements capable of biasing their own inheritance among offspring - and how they behave in formulations that target different regions of the genome. Specifically it looks at a form - split drive - which is a deliberately underpowered version of drive where the gene drive element and its 'motor' are disconnected (not genetically linked). In this case the motor is Cas9 nuclease and works by inducing allelic conversion at a specific target cut site such that a copy of the gene drive is incorporated. This type of split drive has some potential benefits (from one viewpoint at least) in that the invasive force of the drive element is much weaker and therefore easier to physically confine. However by the same token this means its transformative potential is also a lot less. Notwithstanding that, it could be argued that there is increasing agreement that a lot can be learnt from split drive systems and that these can inform, in a more secure way, future releases of more potent gene drives.

There is large body of work within this manuscript. The approach generates split drives with constructs that target 4 separate genes with essential roles in either fertility or general viability. Further, the gene drive element incorporates a recoded version of the target gene that restores function of the target gene while rendering it immune to the cleavage activity of the gene drive. The idea is that this allows one to target a site in the genome that is very conserved (due to functional constraint) and therefore unlikely to be variant in the population or to tolerate alternative alleles that might be generated through Cas9 nuclease activity that is repaired by end-joining. There is a potential added efficacy arising from maternal deposition of nuclease that negatively affects individuals not receiving the drive element and can potentially weed out end-joining alleles. Neither of the ideas are novel per se but this study is a thorough appraisal of how predictions about the behaviour of these types of element in a population actually play out in a population invasion experiment over generations.

I have a few general comments, followed by more specific comments, some of which are listed below while others (very specific) are found on the annotated version of the manuscript which I have uploaded.

This approach - re-coded version of haplosufficient target gene coupled with maternal deposition - as far as I can see it is a good way of improving the robustness of drives with a cargo (most likely to some anti-parasite effector). It still appears to me (correct me if I'm wrong) that the fundamental issue that remains is the propensity for alleles to arise that are both drive-resistant and restore function to the target gene, and the force of selection acting on these will be dependent on the cost associated with the gene drive allele. The deposition effect noted here won't get rid of this particular, problematic allele. The extent to which the accumulation of this type of allele can be retarded by other non-functional end joining mutations that present in the milieu after maternal deposition and prevent the former from having a chance to shine is interesting but I'd like a more explicit treatment of how much this adds. This might be best achieved by comparing it (in thought experiment at least) to the same construct that has zero parental deposition.

I found the choice of figures for inclusion very puzzling. The two best figures in the manuscript are resigned to the supp figures - Fig S2 and Fig S6. Fig S2 contains all the info we need for the performance of the constructs in single generations - it should replace Fig 2 which cherry picks non-comparable split drive combinations. Fig S6 contains some really nice data and should

absolutely be included in the main text. Supp Fig 3 is just a reproduction of a few panels from supp Fig 2 and adds nothing.

There are quite a few sections that are very long and/or contain repetition. I also think that many sections will be difficult for non-experts (and even experts) in gene drive to penetrate. The same goes for the subtlety in some of the design points e.g. the recoding, 3' preference etc. This is a nice practical demonstration of something that was theoretically proposed and there should be more info on this in the methods. An improved supp Fig 1 introduced earlier, or as addition to fig 1, would help.

The modelling appears to this non-expert to be written well and at least allows a general following of the assumptions made. However, at least one of the reviewers should have demonstrable experience in this field. I had a more general question - it would appear that the model outputs, shown on the graph to predict trajectory, actually incorporate estimates of fitness, fecundity and other parameters that were estimated from the cage invasion experiments themselves. So it is not a surprise then that there is good correlation, is it? How would the predicted dynamics compare if estimated only from the single generation crosses and assays?

To explain the observation of under-representation of Cas9 chromosomes, two explanations - chromosome removal after damage and (partial) lethality in the presence of gRNA - are offered. These are reasonable but I wasn't convinced that one is extricable from the other as an explanation

When gRNA and Cas9 are on same chromosome it would be nice to know how distant the sites are and expected recombination frequency between them. This goes to point above - some weird things were seen when both components were on same chromosome - is there a chance that another explanation is related to this proximity? For example if cut site were really close to Cas9 one could get extended resection and conversion that removed the GFP-linked Cas9. Of course that is less likely as the distance between the two increases.

In addition to the under-representation of the Cas9 chromosome in one case (Supp Fig 2Ciii) you also see an over-representation that appears significant, don't you? Any thoughts on this?

Line 243 - is sGD meant here, instead of Cas9 source?

Line 246 - not clear to me why this is 'consistent' with the explanation if it holds for two different sources of Cas9 (albeit on ch III)

Line 248 - to answer the point about deficit of Cas9 chromosomes - is it worth looking in the offspring of the few founders that gave reasonable numbers without the sGD allele and see whether, within this set, the under-representation of GFP-Cas9 chromosomes still holds?

Line 288-289 is it really 7.7% of recipient alleles. To me it seems as though it's 7.7% of all alleles, of which 50% were sGD by inheritance. Therefore shouldn't it be 15.4%?

Line 328 - 329. The logic here, I think, is that because there is a much more restricted set of alleles at *rab5* it implies it is more constrained than the site in *spo11* where there was a mix of NHEJ alleles. Isn't another explanation that there was a clearer 'winner' allele at *rab5* with much higher fitness than others, and therefore selected fast? Whereas at *spo11* there was a milieu of alleles all confer levels of sub-optimal fitness?

Line 371 - why 'surprisingly'? It seems to match the models well - which goes to my query about how much the models are post-fitted versus predictive

Although its inclusion as a means to explain the text is welcome, I struggled with the explanation of Fig 4E (it definitely needs more legend) and, in general, the back and forth invoking of the chromosomal removal vs general lethal mosaicism. It seemed to be argued both ways in places. This may well be my own shortcoming but if I had trouble with it, so will a significant portion of readers.

In supp fig 6 (which I like a lot and should be included in main text) - by eye the ratios of the different EJ alleles between each other is remarkably consistent. Can anything be drawn from this in favouring one or the other of lethal mosaicism vs chromosome removal? Since you have them parsed by presence or absence of sGD would you not expect different ratios. *Note added in review: Actually, reading this again i realise that the left and right column for each timepoint is just a reproduction, with the allele ratios squashed (but unchanged) to represent their overall frequency in the total population. I appreciate the intent but I wonder if it might be less confusing by representing by just leaving the right panel only? I leave my original comment to show you where my confusion arose.

It would be nice to see this approach discussed in context with other approaches that deliberately rely on maternal nuclease deposition for removal of alleles such as toxin:antidote systems etc.

On FigS2 (which should be the main Fig2) it would be very helpful to indicate where differences in inheritance between the sexes are significant for the different chromosomes with the use of asterisks or similar.

More details are needed in some sections of the methods - e.g the deep sequencing of the target sites in the cage experiments is insufficiently explained. e.g the cloning and the cleavage-proofing of the 3'end etc.

I note a peculiar phrase in the Reporting Summary:

"in cage trials where half of the population is used for seeding the next generation, blinding occurred as long as both fly piles displayed similar phenotypic proportions. However, if that was not the case, blinding did not occur to stay true to the total proportions of the cage at a given generation".

This reads a bit clumsily - I'm not any of this is blinding - but what you are saying is that if in the if the frequency was say 0.5 in the scored group and 0.7 in the group for sequencing, you altered the frequency in the seeding group to represent the overall frequency across the two groups?

Reviewer 1

Summary:

The authors present data describing the performance of several Cas9-based split Gene Drives (sGDs) in the model organism *D. melanogaster*. The design is novel in that they target genes essential for viability or fertility and simultaneously introduce recoded versions of the targeted sequences with the homing drive. This creates a system in which undesired NHEJ-repaired alleles are cleared via a dominantly-acting maternal inheritance of active Cas9 that creates lethal/sterile mosaics in the next generation.

The split-drive architecture allowed for a combinatorial generation of trans-heterozygote 'master strains' to empirically test many different design variations. The authors do a good job of highlighting the nuanced ways in which genetic context impact the performance of specific drives throughout the results and discussion. These genetic context effects were not examined mechanistically in this paper (i.e. with careful measurements of Cas9 expression from different chromosomal loci) but this does not detract from the overall message of the paper.

This study is well-designed, well-communicated, and complete. The authors provide an end-to-end report of their work that includes system design, construction and testing, population-level experiments, and modeling. Their conclusions are supported by ample experimental data. I think this paper will be well-received by the field.

There are suggested improvements listed beneath my signature. I recommend this paper for publication in Nature Communications with minor revisions (i.e. no additional experiments). **I welcome questions or concerns about the content of this review by the authors of the manuscript or editors at Nature Biotechnology.** If it is against editorial policy to have non-anonymous reviews, please remove my name below (although I will note that I include my contact information in all peer-reviews and it has only led to neutral or positive outcomes so far).

Substantive points to consider during revision:

1. *The readability of the paper could be improved to make this study more accessible. Several specific examples are given in the minor points section below. Two common themes would be to (i) be more concise with the description of the results. The current word count is probably ~60% over the Nature Communications suggested limit of 5,000 words. And (ii) the figure legends often contain much more information than is needed to understand the figure. I suggest not duplicating statements about the results both in the main text and in figure legends.*

We have revised the text as suggested to be as concise as possible and have reduced the word count substantially.

2. *The stochastic implementation of the mathematical model (i.e. Figure 5) routinely underestimates the drive efficiency for the first few generations of the simulations. It is notable that 3 of 3 experimental results appear as outliers to the 100 simulations in panels A and C, and 3 of 4 as outliers in panel F. While the experimental results in panel B do not reach the status of*

being outliers, the same trend seems to exist wherein the model is underestimating early drive efficiency. The authors' description of their modeling approach is sound, but the paper would be improved if the authors provided more text to explain this observation.

This is a good point and we have considered potential reasons for the better than expected performance of the drives predicted by the modeling based on the single-generation cross data. One plausible possibility, is that sGD-bearing females are mating more efficiently than males. Since transmission of the drive via females sGD parents is typically ~15% greater than from males, it could result in the observed transient overperformance relative to the modeling which assumes equal mating success of sGD males and females. Indeed, the divergence between experimental and predicted trajectories is much less for the *proalpha2* drive which display little sex-specific difference in transmission in single generation crosses. We added this potential explanation to the revised text.

- 3. As a primary driver for this study seems to be the impact this system has on non-functional NHEJ alleles, it would be good for the authors to revisit the issue of overcoming genetic resistance in the context of real-world biocontrol scenarios. How low will resistant rates need to be for a field application? How does the rate of resistant alleles compare in this system versus other published systems? Are aspects of this system expected to work synergistically with other approaches (e.g. multiplexed guides)? The authors do this in part in the last paragraph, but it is difficult to glean the strengths and weaknesses of this drive design versus others.*

This is a tough question to answer based on existing information since it depends on several unknown factors including: **1)** how well will drive performs in nature compared to the lab. This problem is particularly difficult to assess for mosquitoes that display swarming behaviors in which males must compete under stringent conditions for mating; **2)** what levels of transgene factors might be required to achieve desired biological efficacy (e.g., would it require only a single copy of an anti-malarial effector cassette or two?), and **3)** the relative fitness costs/stability of gene-drives in natural populations. In the cage experiments presented in this study we did not detect any significant fitness costs associated with the drive element alone, but whether that would also be true in a natural context is hard to gauge. Another question is whether the full-drive configuration might have differing associated fitness costs than the split-drive. As mentioned in the manuscript, we have generated a "hacked" *spo11* line with a full drive configuration and although its initial rate of increase is less than that observed for split drive (perhaps because the drive cannot separate from the Cas9 fitness cost during the drive process), it nonetheless achieves the same final level of introgression, suggesting that there is also no significant post-drive fitness cost associated the full-drive configuration, at least in the lab. Since this is a lengthy point, we suggest that it might be better to give it proper consideration in the context of a review.

- 4. Provision of a recoded version of the 3' region of the target gene CDS is an important aspect of this paper, yet no detail is given to the design of this recoded CDS. More details should be included, and possibly sequence comparisons of the wildtype CDS and recoded CDS in the SI. For example, is just the sgRNA target site recoded, or the whole fragment? How were variant codons assigned and what is the expected impact on translational elongation?*

We have added more information to the supplementary information detailing the nature of the recoding which we do over the entire length of the cDNA C-terminal region.

Minor points:

1. Line 69: you have two spaces after the word 'resistant'
We have deleted the extra space.
2. Line 84: have -> has
We have corrected this typo.
3. Line 243: "Carrying both the Cas9 transgene and Cas9 source". I am not sure what is meant by this sentence. What is a Cas9 source, if not the cas9 transgene?
We had meant to state the sGD element and Cas9 source and have made this correction.
4. Line 300: delete the word 'unaltered'. It's inclusion indicates that there is such a thing as an 'altered WT allele', which would no longer be WT.
We agree and have deleted the word unaltered as suggested.
5. Line 401: Sentence needs to be revised somehow. Possibly to: "...given the remaining WT alleles present..."
We have rephrased this awkward passage as: "...we wondered whether we could increase its level of introgression in population cage experiments given that nearly all non-drive alleles that remained following elimination of the Cas9 source were WT."
6. Line 437: "through"
We have corrected this typo.
7. Line 655: correct in-text citation to a number corresponding with the list of references.
We have added the Gloor reference.
8. Line 665: Add a space between "18 h"
We have corrected this typo.
9. Figure 1b: the graphical difference between the 'different chromosome' and 'same chromosome' is quite subtle. I suggest adding a semicolon between the two chromosomes at left, which is compatible with typical genotype representations, would make the difference between the left and right drawings more obvious.
We have followed this suggestion to clarify the figure.
10. Figure 2: be consistent with use of significant figures. In the Mean Inheritance row (top) some experiments are reported with two significant figures, and others with three.
We have revised the percentages to consistently present three significance figures.
11. Figure 2 Legend: All text after "Super-Mendelian..." is a description of the results, not of the figure, and that text is better moved to the results section of the main text (Although it is probably already stated there, so can just be deleted).
We have followed this suggestion and deleted this redundant text from the legend.
12. Figure 3: Panel labels A,B,C,D are disproportionately large.

We have reduced the size of the labeling font accordingly. We also moved supplementary data from the cage sequencing to this main figure as suggested by reviewer 2 and have adjusted the labeling accordingly to be consistent across all panels.

13. Figure 3 Legend: The sentence beginning “Each section..” is redundant to information earlier in the legend and can be removed.

We have followed this suggestion and deleted the redundant passage.

14. Figure 4E: This part of the figure is difficult to interpret. The inset at top left is too small to see well, and the yellow dots in the equivalent figures at right are difficult to see. It is unclear why the word ‘and’ in the top example and the word ‘but’ in the bottom two examples. In general, I like the idea behind this panel (i.e. providing graphical explanation behind the different rates of loss of the Cas9 cassette), but unfortunately the execution of this panel is still a bit confusing.

We have revised and simplified this graphical summary to clarify these points.

15. Figure 4 Legend: There is a lot of text in the legend that is describing the results and would be better placed in the main text results section instead of the figure legend.

We have trimmed down the legend to the essential material points. Since we also needed to shorten the text substantially, we hope this compaction has not compromised the clarity of our presentation of the data.

16. Figure 1,2,4,5: Use of red-green as the primary differentiation between data sets is not ideal for audience with red-green color blindness (8% of males of northern European descent).

We appreciate this important point and have changed our color scheme to being dichromatically friendly.

Reviewer #2 (Remarks to the Author):

This manuscript looks at gene drives - selfish genetic elements capable of biasing their own inheritance among offspring - and how they behave in formulations that target different regions of the genome. Specifically it looks at a form - split drive - which is a deliberately underpowered version of drive where the gene drive element and its 'motor' are disconnected (not genetically linked). In this case the motor is Cas9 nuclease and works by inducing allelic conversion at a specific target cut site such that a copy of the gene drive is incorporated. This type of split drive has some potential benefits (from one viewpoint at least) in that the invasive force of the drive element is much weaker and therefore easier to physically confine. However by the same token this means its transformative potential is also a lot less. Notwithstanding that, it could be argued that there is increasing agreement that a lot can be learnt from split drive systems and that these can inform, in a more secure way, future releases of more potent gene drives.

There is large body of work within this manuscript. The approach generates split drives with constructs that target 4 separate genes with essential roles in either fertility or general viability. Further, the gene drive element incorporates a recoded version of the target gene that restores function of the target gene while rendering it immune to the cleavage activity of the gene drive. The idea is that this allows one to target a site in the genome that is very conserved (due to functional constraint) and therefore unlikely to be variant in the population or to tolerate alternative alleles that might be generated through Cas9 nuclease activity that is

repaired by end-joining. There is a potential added efficacy arising from maternal deposition of nuclease that negatively affects individuals not receiving the drive element and can potentially weed out end-joining alleles. Neither of the ideas are novel per se but this study is a thorough appraisal of how predictions about the behaviour of these types of element in a population actually play out in a population invasion experiment over generations.

I have a few general comments, followed by more specific comments, some of which are listed below while others (very specific) are found on the annotated version of the manuscript which I have uploaded.

This approach - re-coded version of haplosufficient target gene coupled with maternal deposition - as far as I can see it is a good way of improving the robustness of drives with a cargo (most likely to some anti-parasite effector). It still appears to me (correct me if I'm wrong) that the fundamental issue that remains is the propensity for alleles to arise that are both drive-resistant and restore function to the target gene, and the force of selection acting on these will be dependent on the cost associated with the gene drive allele. The deposition effect noted here won't get rid of this particular, problematic allele. The extent to which the accumulation of this type of allele can be retarded by other non-functional end joining mutations that present in the milieu after maternal deposition and prevent the former from having a chance to shine is interesting but I'd like a more explicit treatment of how much this adds. This might be best achieved by comparing it (in thought experiment at least) to the same construct that has zero parental deposition.

The reviewer raises two important points. First, they correctly point out that lethal mosaicism, even in its most potent form, cannot eliminate functional in-frame alleles as indeed revealed in several of the cage experiments in which only in-frame and presumably functional NHEJ alleles (in some case verified to be so) remain at the end of the drive trial. None of these alleles competitively reduce the prevalence of the recoded drive, however, which is very encouraging since such a wide array of such alleles were generated in the different loci, which suggests that the recoded target gene expressed under control of endogenous regulatory sequences have little if any fitness costs associated with them (i.e., they persist undiminished over 20 generations long after the Cas9 transgene has dropped out of the population). As we point out, the best strategy is to target coding sequences that greatly reduce the generation of such alleles (e.g., single amino acid codons such as those for tryptophan or methionine - or cutting between the first and second nucleotide of a codon for an essential catalytic residue).

The second question is a bit more difficult to answer, namely if one could somehow reduce the maternal deposition (e.g., using a cleaner germline-specific promoter), would this lead to better drive outcomes (i.e., greater final prevalence of the drive). We have not modeled this question extensively, but it is likely that there will be tradeoffs. On the one hand, reducing the amount of maternal Cas9/gRNA mutagenesis of the target gene will lead to fewer NHEJ alleles being generated and hence to a higher ratio of HDR to NHEJ editing outcomes. On the other hand, however, the intensity of lethal or sterile mosaicism will be reduced. While we fully agree that this is a very interesting and important question to consider further, since this is not a trivial point to discuss, it may be more appropriate for a review where more space could be granted to the question.

I found the choice of figures for inclusion very puzzling. The two best figures in the

manuscript are resigned to the supp figures - Fig S2 and Fig S6. Fig S2 contains all the info we need for the performance of the constructs in single generations - it should replace Fig 2 which cherry picks non-comparable split drive combinations. Fig S6 contains some really nice data and should absolutely be included in the main text. Supp Fig 3 is just a reproduction of a few panels from supp Fig 2 and adds nothing.

We agree with the reviewer on these excellent points and accordingly, we have moved the two relevant supplementary sections into revised versions of Fig. 2 (comprehensive single-generation crossing data) and Fig. 3 (deep sequencing of analysis from cage studies). In the case of the revised Fig. 2, we emphasize the key data presented in the current Fig. 2 (e.g., by using larger graphing symbols) to aid the reader in spotting these results as they are discussed further in the text.

There are quite a few sections that are very long and/or contain repetition. I also think that many sections will be difficult for non-experts (and even experts) in gene drive to penetrate. The same goes for the subtlety in some of the design points e.g. the recoding, 3' preference etc. This is a nice practical demonstration of something that was theoretically proposed and there should be more info on this in the methods. An improved supp Fig 1 introduced earlier, or as addition to fig 1, would help.

We have followed this recommendation and have eliminated redundancy throughout the text and reduce the total number words.

The modelling appears to this non-expert to be written well and at least allows a general following of the assumptions made. However, at least one of the reviewers should have demonstrable experience in this field. I had a more general question - it would appear that the model outputs, shown on the graph to predict trajectory, actually incorporate estimates of fitness, fecundity and other parameters that were estimated from the cage invasion experiments themselves. So it is not a surprise then that there is good correlation, is it? How would the predicted dynamics compare if estimated only from the single generation crosses and assays?

The parameter ranges explored in the modeling do indeed come from results of the single generation crosses as the reviewer expects they should. We have clarified this point in the revised text.

To explain the observation of under-representation of Cas9 chromosomes, two explanations - chromosome removal after damage and (partial) lethality in the presence of gRNA - are offered. These are reasonable but I wasn't convinced that one is extricable from the other as an explanation.

The two effects reducing prevalence of the Cas9 chromosome can be estimated by using data from single generation crosses to assess directly the damage effect (only observed for the *prosalpha2* sGD drive using the vCas9-III and nosCas9-III lines) and from their behavior in cage trials in which the frequency of Cas9 declined only when combined with particular specific sGD drives (e.g., *prosalpha2*, *rab5*, and *spo11* drives but not the *rab11* drive). Modeling of the cage experiments allowed estimates of these latter drive+Cas9 fitness costs based on the rate of decline in Cas9 frequency, which we suggest could be explained by partial haploinsufficiency for these loci. We have clarified this reasoning in the revised text.

When gRNA and Cas9 are on same chromosome it would be nice to know how distant the sites are and expected recombination frequency between them. This goes to point above - some weird things were seen when both components were on same chromosome - is there a chance that another explanation is related to this proximity? For example if cut site were really close to Cas9 one could get extended resection and conversion that removed the GFP-linked Cas9. Of course that is less likely as the distance between the two increases.

The vCas9-III source is quite close in terms of recombination distance from the *prosalpha2* gRNA cleavage site (~2 centimorgans) but, as the reviewer suspects, these megabase distances are quite distant with regard to the extent of 3' resection during HDR (in flies resection distances measured in other published studies average a few hundred bases extending occasionally to the range of a kilobase). So, we do not expect that local gene conversion events would be responsible for the observed reductions in Cas9 frequency.

In addition to the under-representation of the Cas9 chromosome in one case (Supp Fig 2Ciii) you also see an over-representation that appears significant, don't you? Any thoughts on this?

The reviewer is correct that there is a modest excess in the average inheritance of the nosCas9-II source in the *rab11* sGD experiments (new Fig. 2C). There is no obvious explanation for why an unlinked source of Cas9 particularly for this sGD, which exhibits little if any fitness cost either even in combination with Cas9 (e.g., Fig. 4C). Since there is a fair amount of variation in this single generation cross, we believe that this modest deviation from expected Mendelian inheritance most likely is due to stochastic variation. We have nonetheless added significance stars in Fig. 2C to indicate this deviation in this particular *rab11* sGD experiment (we similarly indicated instances of sub-Mendelian inheritance of Cas9 in Fig. 2D).

Line 243 - is sGD meant here, instead of Cas9 source?

Yes, and we have made this correction.

Line 246 - not clear to me why this is 'consistent' with the explanation if it holds for two different sources of Cas9 (albeit on ch III).

In the case of the nosCas9-III *prosalpha2* sGD single generation crosses (Fig. 2D), we note that there is a greater reduction in the master-female crosses than in the master-male crosses (this is also true for vCas9-III). If target chromosome damage were the source of this reduction (which it is likely to be for the closely linked vCas9-III element) then result should be different for the nCas-III source (less prevalent in master-female versus male crosses). Thus, we interpret these results as evidence for an intrinsic factor leading to reduced nosCas9-III inheritance in females such as some form of greater Cas9 activity. We have revised the sentence in question to clarify this point as follows:

*“Selective reduction of the Cas9 element in master female crosses employing the 3rd chromosome nCas9-III (Fig. 2d) is consistent with a greater relative activity of this Cas9 source since it is unlinked to the *prosalpha2* sGD locus and should freely recombine with any chromosomal damage incurred at the gRNA cut site.”*

Line 248 - to answer the point about deficit of Cas9 chromosomes - is it worth looking in the offspring of the few founders that gave reasonable numbers without the sGD allele and see whether, within this set, the under-representation of GFP-Cas9 chromosomes still holds?

That is a clever idea. However, when we checked, there are not a sufficient number of such vials to perform this analysis in a statistically significant fashion.

Line 288-289 is it really 7.7% of recipient alleles. To me it seems as though it's 7.7% of all alleles, of which 50% were sGD by inheritance. Therefore shouldn't it be 15.4%?

The reviewer is correct. We have deleted the term recipient and rephrased this point as "7.7% of total F₂ alleles" to be consistent with other allelic comparisons and interpretation of the figure graph panel.

Line 328 - 329. The logic here, I think, is that because there is a much more restricted set of alleles at rab5 it implies it is more constrained than the site in spo11 where there was a mix of NHEJ alleles. Isn't another explanation that there was a clearer 'winner' allele at rab5 with much higher fitness than others, and therefore selected fast? Whereas at spo11 there was a milieu of alleles all confer levels of sub-optimal fitness?

Perhaps this is a bit of a semantic point. What we had intended to imply is that there appears to be a greater constraint in stably transmittable NHEJ alleles generated by the *rab5* sGD relative to the *spo11* sGD. It also seems that there may be different spectra of mutant alleles generated at the two loci, suggested by relatively lower fraction of NHEJ alleles that are recovered in single *rab5* generation crosses and the lower complexity of such alleles observed at generation 2 for *rab5* relative to *spo11*. We have clarified these potential factors by adding the following sentence:

"Alternatively, particular rab5 alleles may become fixed in a biased fashion by outcompeting other NHEJ events that carry greater fitness costs."

Line 371 - why 'surprisingly'? It seems to match the models well - which goes to my query about how much the models are post-fitted versus predictive.

We were surprised by these results because among the single generation crosses, the *prosalpha2* sGD displayed the best drive performance in both females and males, where it was by far the best. Yet, using the same Cas9 strain in cages the level of drive introgression was the lowest. This was the pivotal result that made us consider the various roles of Cas9 chromosome killing and substantial selection against individuals carrying both the sGD and Cas9 (perhaps due to moderate locus haplo-insufficiency). Indeed, it was through iterative modeling that we arrived at our current hypothesis regarding the multiple effects at play, which then lead to our performing the definitive experiment of testing the *prosalpha2* sGD drive in the context of 100% Cas9 prevalence, where we found that the sGD drove rapidly to complete introgression. We have attempted to clarify this point with the following revised phrasing:

"This population level behavior was surprising as it contrasted markedly with the efficient transmission of this sGD through both males and females in single-generation crosses using the same vCas9-III source (>99%)."

Although its inclusion as a means to explain the text is welcome, I struggled with the

explanation of Fig 4E (it definitely needs more legend) and, in general, the back and forth invoking of the chromosomal removal vs general lethal mosaicism. It seemed to be argued both ways in places. This may well be my own shortcoming but if I had trouble with it, so will a significant portion of readers.

Reviewer 1 made a similar comment which we have addressed by simplifying and revising the phrasing of the summary scheme to make it more accessible and helpful as graphic support.

*In supp fig 6 (which I like a lot and should be included in main text) - by eye the ratios of the different EJ alleles between each other is remarkably consistent. Can anything be drawn from this in favouring one or the other of lethal mosaicism vs chromosome removal? Since you have them parsed by presence or absence of sGD would you not expect different ratios. *Note added in review: Actually, reading this again I realize that the left and right column for each timepoint is just a reproduction, with the allele ratios squashed (but unchanged) to represent their overall frequency in the total population. I appreciate the intent but I wonder if it might be less confusing by representing by just leaving the right panel only? I leave my original comment to show you where my confusion arose.*

We have added labels at the top of the column depicting the distribution of all (total alleles) to help clarify this understandable confusion. These panels have been moved into the main Fig. 3 as also recommended by this reviewer.

It would be nice to see this approach discussed in context with other approaches that deliberately rely on maternal nuclease deposition for removal of alleles such as toxin:antidote systems etc.

We have added mention of toxin:antidote (e.g., Medea systems) in this context.

On FigS2 (which should be the main Fig2) it would be very helpful to indicate where differences in inheritance between the sexes are significant for the different chromosomes with the use of asterisks or similar.

We have followed this suggestion (we also moved the supplemental data from Fig. S2 into main Fig. 2 following this reviewer's recommendation).

More details are needed in some sections of the methods - e.g. the deep sequencing of the target sites in the cage experiments is insufficiently explained. e.g. the cloning and the cleavage-proofing of the 3'end etc.

We have added this suggested information to the revised Materials and Methods section.

I note a peculiar phrase in the Reporting Summary: "in cage trials where half of the population is used for seeding the next generation, blinding occurred as long as both fly piles displayed similar phenotypic proportions. However, if that was not the case, blinding did not occur to stay true to the total proportions of the cage at a given generation". This reads a bit clumsily - I'm not any of this is blinding - but what you are saying is that if in the if the frequency was say 0.5 in the scored group and 0.7 in the group for sequencing, you altered the frequency in the seeding group to represent the overall frequency across the two groups?

We will modify the Reporting Summary once we can access it to rephrase and clarify this sentence. The reviewer is correct: If the frequencies scored 'blindly', or 'randomly', for seeding and sequencing groups were distant, they were altered to represent the overall frequency more accurately to pass onto the next generation.

Reviewers' Comments:

Reviewer #1:

Remarks to the Author:

Dear editors and authors,

I have reviewed the revised manuscript, including the responses to both reviewers, and feel that the reviewer comments were adequately addressed. This is an impressive piece of work and I am excited to see it in print.

Reviewer #2:

Remarks to the Author:

I think the manuscript is much improved and am happy that the authors have satisfactorily responded to my queries.

I have a few minor points below, but I do not need to see the manuscript again.

Re: the 'surprising results' relating to α_2 construct. This is well argued in the response to review and the point made forcefully - arguably better so than in the main text (one might consider transferring some of style of this response into discussion of results. This is a decision of style, feel free to take it or leave it)

Introgression probably means different things to different people - for example increase the 'level of introgression' to some people may have the limited definition of reduce the amount of linked sequence that accompanies the gene drive. Whereas you are talking about biased inheritance, generation on generation. Admittedly this probably does lead to more and more introgression (in the above sense) each time but would it not be better to use the term 'drive' here?

Can we use another word than 'hacking' for the secondary modification of the non-autonomous drive constructs to autonomous constructs? It has all sorts of negative connotations that are not helpful.

The modifications to the figures have helped - in response to the other reviewer's suggestion, the space around the semi-colon on Fig 2 should be made slightly larger to show that that they are separate chromosomes.

re: Reporting Summary - it appears the average of frequencies scored from the seeding and sequencing groups was used. That info should go in the methods. Presumably this could have the effect on reducing stochastic extremes that one might observe if one was only one of those groups.

Reviewer #1 (Remarks to the Author): (second revision)

Dear editors and authors,

I have reviewed the revised manuscript, including the responses to both reviewers, and feel that the reviewer comments were adequately addressed. This is an impressive piece of work and I am excited to see it in print.

Reviewer #2 (Remarks to the Author): (second revision)

I think the manuscript is much improved and am happy that the authors have satisfactorily responded to my queries.

I have a few minor points below, but I do not need to see the manuscript again.

Re: the 'surprising results' relating to prosalpha 2 construct. This is well argued in the response to review and the point made forcefully - arguably better so than in the main text (one might consider transferring some of style of this response into discussion of results. This is a decision of style, feel free to take it or leave it)

We modified the text slightly, adding a couple sentences that may make it clearer (added parts in blue): This population-level behavior was surprising as it contrasted markedly the efficient transmission of this sGD through both males and females in single-generation crosses, **where it displayed the greatest drive potential**, using the vCas9-III source (>99%). **Yet, using the same Cas9 strain in cages, this sGD achieved only a modest level of introduction (Fig. 4d, thin dashed lines).**

Introgression probably means different things to different people - for example increase the 'level of introgression' to some people may have the limited definition of reduce the amount of linked sequence that accompanies the gene drive. Whereas you are talking about biased inheritance, generation on generation. Admittedly this probably does lead to more and more introgression (in the above sense) each time but would it not be better to use the term 'drive' here?

Introgression was replaced by 'introduction' or 'spread', depending on the context.

Can we use another word than 'hacking' for the secondary modification of the non-autonomous drive constructs to autonomous constructs? It has all sorts of negative connotations that are not helpful.

We removed the word hacking in one of the instances (introduction) and changed it from 'hacking' to 'autonomous drives' in the other (discussion). Also, in the discussion, we removed the HACK acronym and only left "homology assisted CRISPR knock-in".

The modifications to the figures have helped - in response to the other reviewer's suggestion, the space around the semi-colon on Fig 2 should be made slightly larger to show that that they are separate chromosomes.

We increased the font of the semicolon from 21 to 24 and spaced out the chromosomes in Figure 1, which is the figure that the reviewer refers to (not Fig.2).

re: Reporting Summary - it appears the average of frequencies scored from the seeding and sequencing groups was used. That info should go in the methods. Presumably this could have the effect on reducing stochastic extremes that one might observe if one was only one of those groups.

We included the above comment in the methods section as well: 'If the two pools differed much phenotypically, frequencies were averaged in order to reduce variability and stochastic extremes'.